# PICS: Pairwise Image Compositing with Spatial Interactions

**Hang Zhou**♣    **Xinxin Zuo**◇    **Sen Wang**◇    **Li Cheng**♣
♣University of Alberta    ◇Concordia University

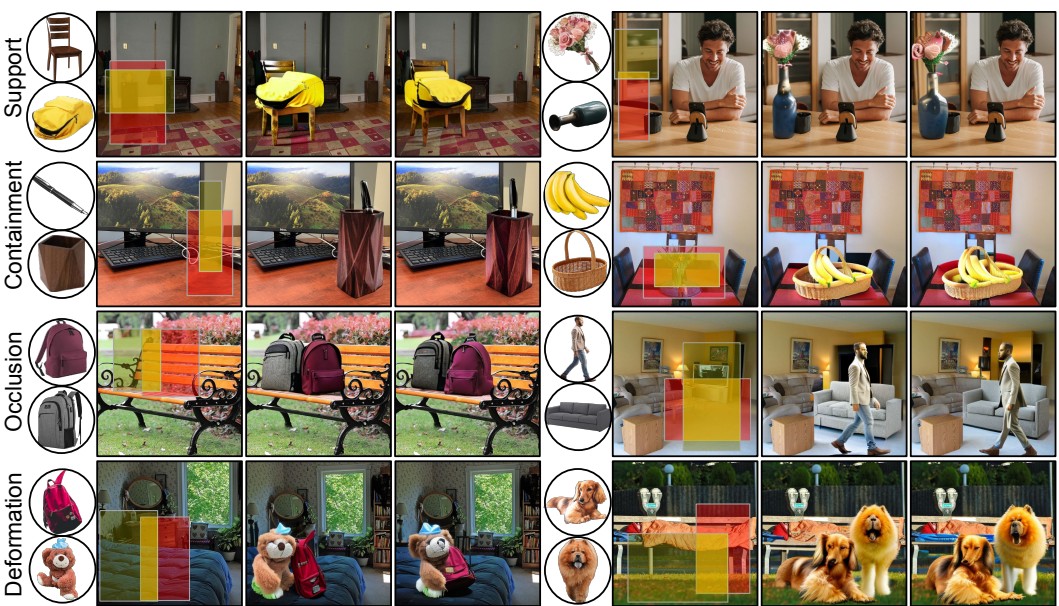

Figure 1: Our method generates spatially plausible and visually realistic pairwise compositions. Each row illustrates two examples, consisting of (from left to right) the objects, the masked background, and two exemplar composite results. Additional comparative results appear in the appendix.

## ABSTRACT

Despite strong single-turn performance, diffusion-based image compositing often struggles to preserve coherent spatial relations in pairwise or sequential edits, where subsequent insertions may overwrite previously generated content and disrupt physical consistency. We introduce *PICS*, a self-supervised composition-by-decomposition paradigm that composes objects *in parallel* while explicitly modeling the *compositional interactions* among (fully-/partially-)visible objects and background. At its core, an Interaction Transformer employs mask-guided Mixture-of-Experts to route background, exclusive, and overlap regions to dedicated experts, with an *adaptive* $\alpha$-blending strategy that infers a compatibility-aware fusion of overlapping objects while preserving boundary fidelity. To further enhance robustness to geometric variations, we incorporate geometry-aware augmentations covering both out-of-plane and in-plane pose changes of objects. Our method delivers superior pairwise compositing quality and substantially improved stability, with extensive evaluations across virtual try-on, indoor, and street scene settings showing consistent gains over state-of-the-art baselines. Code and data are available at `https://github.com/RyanHangZhou/PICS`

## 1 INTRODUCTION

The purpose of image compositing is to seamlessly integrate objects or regions, sourced from different images, into a unified and visually plausible image. This fundamental task has recently garnered

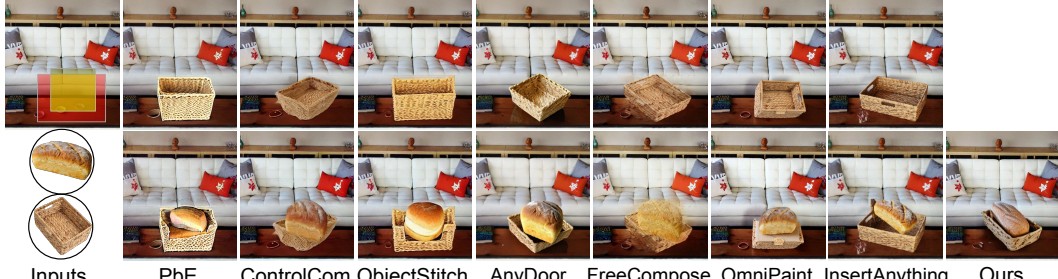

Figure 2: Visual comparison of pairwise support relations across Paint-by-Paint, ControlCom, ObjectStitch, AnyDoor, FreeCompose, OmniPaint and InsertAnything. Left: backgrounds and two objects; right: compositing results. The first row shows composites with the basket, and the second row shows subsequent composites obtained by adding the bread on top. Unlike prior methods that suffer from contact artifacts and fidelity loss, our approach performs parallel compositing, effectively handling spatial occlusions and yielding consistent results with preserved fine-grained structure.

considerable attention, particularly in film production and photo retouching, where it facilitates the seamless blending of diverse visual elements (Mortensen & Barrett, 1995). In the film industry, advanced compositing techniques, often coupled with digital manipulation, enable the realistic integration of vintage footage into modern scenes (Brinkmann, 2008; Wright, 2017).

Earlier compositing methods, including image blending (Smith & Blinn, 1996; Pérez et al., 2003), harmonization (Tsai et al., 2017; Guerreiro et al., 2023), and GAN-based models (Chen & Kae, 2019; Azadi et al., 2020), refine the appearance of inserted regions but generalize poorly across diverse backgrounds. Recent diffusion models (Ho et al., 2020; Dhariwal & Nichol, 2021) offer stronger generative capability and flexible conditioning, substantially advancing image compositing. Building on this, several approaches encode objects as visual prompts, enabling diffusion-based compositing across varied contexts (Song et al., 2023; Yang et al., 2023; Lu et al., 2023; Chen et al., 2024b; Song et al., 2024; Canet Tarrés et al., 2024; Chen et al., 2024c; 2025; Tian et al., 2025; Yu et al., 2025; Song et al., 2026). Despite these gains, such methods remain vulnerable in *multi-turn*[1] settings: sequential compositing often disrupt prior content, degrading compositional consistency and visual fidelity, as shown in Figure 2, particularly regarding realistic object interactions.

We posit that instability in multi-turn compositing arises from the lack of explicit modeling of object-object interactions. In real-world scenes, objects rarely occur in isolation; fundamental pairwise relations such as support (Jiang et al., 2012), containment (Shamsian et al., 2020), occlusion (Lazarow et al., 2020), and deformation (Romero et al., 2022) structure spatial plausibility. These relations define the basic unit for compositional reasoning (Patel et al., 2024; Mishra et al., 2025), enabling systematic evaluation of the limitations of existing diffusion-based compositing methods.

To address these challenges, we introduce PICS, a *parallel* image compositing model that performs pairwise compositing in a single pass while preserving both object-object and object-background consistency. Built on a latent diffusion backbone with ControlNet conditioning on the masked background, PICS employs *Interaction Transformer Blocks* with *mask-guided Mixture-of-Experts* (MoE): background, per-object exclusive regions, and overlaps are deterministically routed to dedicated experts. The background expert is identity-preserving; exclusive-region experts apply cross-attention from scene to individual object; and the overlap expert employs an adaptive attention-gated $\alpha$-blending strategy that dynamically mediates object presence conditioned on background, yielding spatially coherent interactions in the intersection region. Additionally, we incorporate *geometry-aware augmentations* to handle both in-plane and out-of-plane pose variations of objects.

Our contributions are summarized as follows:

**Parallel Compositing.** By modeling pairwise image compositing in parallel, our approach effectively avoids the artifacts inherent to step-wise compositing.

**Interaction Transformer Block.** We propose mask-guided Mixture-of-Experts for region-aware modeling, together with an adaptive $\alpha$-blending module that achieves boundary-consistent and spatially coherent pairwise composites.

---

[1]We use "turn" to denote a composition or editing round, to avoid confusion with diffusion sampling steps.

**Comprehensive Evaluation.** Extensive experiments demonstrate that PICS significantly improves pairwise compositing quality across various scenarios.

## 2 RELATED WORK

**Image compositing.** Image compositing is the task of inserting an image-based object into a background image while maintaining visual and contextual consistency. Early approaches fall into three categories: image blending, which focuses on smoothing boundaries for seamless transition between the inserted object and the background (Smith & Blinn, 1996; Pérez et al., 2003), image harmonization, which adjusts color and illumination to achieve visual compatibility (Tsai et al., 2017; Guerreiro et al., 2023; Chen et al., 2024c), and GAN-based models (Chen & Kae, 2019; Zhan et al., 2019; Azadi et al., 2020), which targets geometry consistency by adversarial training. Recent work represents objects as visual prompts and conditions diffusion models (Ho et al., 2020; Dhariwal & Nichol, 2021) to support more general, adaptive compositing (Song et al., 2023; Yang et al., 2023; Lu et al., 2023; Chen et al., 2024b; Song et al., 2024; Canet Tarrés et al., 2024; Chen et al., 2024c; 2025; Tian et al., 2025; Yu et al., 2025; Song et al., 2026). Despite these advances, most frameworks remain essentially *single-turn*: each composite is generated from a single prompt, without support for iterative composition. In complex scenes where multiple objects are added sequentially and may overlap or contact (Zhan et al., 2024; Ao et al., 2025; Liu et al., 2025), models trained only on foreground-background pairs often produce artifacts, especially near overlaps and contacts, due to the absence of explicit object-object relation modeling. A related line, multi-object image customization, personalizes images with multiple objects by jointly generating foreground and background layouts simultaneously (Bao et al., 2024; Chefer et al., 2023; Dahary et al., 2024; Gu et al., 2023; Wang et al., 2024). In contrast, we directly target pairwise object compositing, yielding spatially coherent and visually faithful composition.

**Multi-turn image editing.** Diffusion models have significantly advanced image editing, producing results that are both realistic and diverse. However, most methods operate in a *single-turn* regime: each edit is generated from an isolated prompt without carrying state across rounds (Saharia et al., 2022; Chen et al., 2024a; Cai et al., 2025). As a result, they preserve local fidelity but struggle to maintain *global* coherence over a sequence of edits. To address this, recent work introduces *multi-turn* editing that conditions each round on prior outputs and instructions (Zhou et al., 2025; Gupta et al., 2025; Avrahami et al., 2025), which aligns with our setting: each new instruction must respect previously composed content and preserve cross-turn consistency. This dependency on past edits, in turn, makes the task prone to error propagation and semantic drift, an issue analogous to multi-turn dialogue in language models (Wang et al., 2018; Kwan et al., 2024; Duan et al., 2024; Laban et al., 2025). Notably, sustaining coherence in practice hinges on how compositions handle *pairwise* object interactions. Without explicit modeling, methods that perform well in single-turn settings often fail to manage occlusions and preserve boundary consistency, as illustrated in Figure 5.

**Projected object relations.** A 3D scene encodes rich spatial relations among objects that, once rendered to 2D, appear as *projected* interactions between instances. For example, 2D occlusion arises from 3D depth ordering, while support and containment persist through contact and enclosure cues. Prior work has leveraged such projected relations for scene understanding, compositional reasoning, and image synthesis. Building on this perspective, we study how explicitly modeling these relations yields more realistic and spatially consistent 2D object compositing. Reasoning about occluded objects is a long-standing challenge in spatial understanding. Occlusion-annotated datasets (Martin et al., 2001; Zhu et al., 2017; Zhan et al., 2024) and self-supervised approaches (Zhan et al., 2020) establish the foundations for occlusion handling from a perceptual standpoint, and subsequent methods further advance amodal completion/de-occlusion (Ling et al., 2020; Ke et al., 2021; Zhou et al., 2021; Liu et al., 2024). On the generative side, LaRender (Zhan & Liu, 2025) introduces explicit occlusion control in image generation.

## 3 METHODOLOGY

We begin with the parallel pairwise image compositing pipeline in Subsection 3.1, followed by the interaction transformer in Subsection 3.2 that models interactions among objects and the background. Finally, we introduce two geometry-aware augmentations in Subsection 3.3.

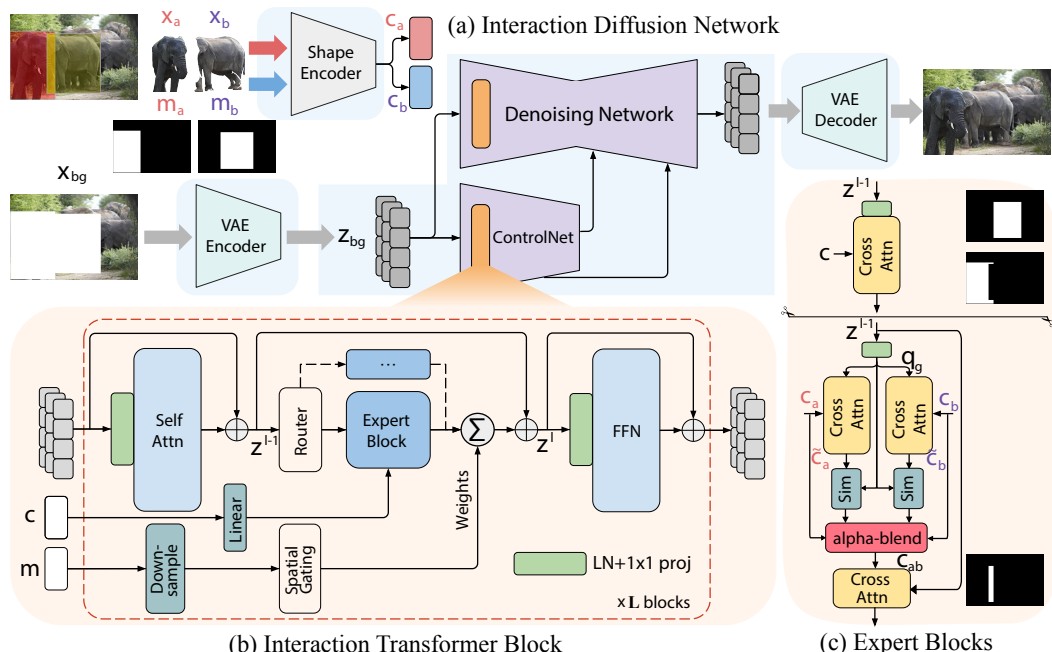

Figure 3: Overview of PICS. Input data are constructed by decomposing the target image into a background and pairwise objects with their designated regions. (a) The interaction diffusion network composites the objects into the background. (b) The interaction transformer block, shared across both branches, models interactions among objects and with the background. (c) Expert blocks focus on distinct spatial regions. All notations are defined in the main text for clarity.

## 3.1 PAIRWISE IMAGE COMPOSITING

**Exploring two-turn compositing.** Object-to-object contact is a pervasive phenomenon in the physical world. When a 3D scene is projected onto a 2D image, objects tend to (partially-)occlude each other, leading to what we term interdependent objects. This poses a central challenge for pairwise image compositing: *how can synthesized images with occlusions remain visually realistic, and to what extent do existing methods effectively model such occlusion and spatial interactions?* To investigate this, we systematically examine the strengths and weaknesses of current single-object compositing approaches when extended to scenarios where inserted objects interact spatially.

A straightforward baseline composes objects in sequence. For the compositing order, we adopt the classical Painter's Algorithm (Newell et al., 1972): objects are ranked by a depth proxy, estimated from the vertical position of their 2D bounding boxes, and composited farther first, nearer last, ensuring that later insertions occlude earlier ones. In Figure 2, existing methods often degrade at interaction boundaries, largely due to foreground-background partitioning in data construction that ignores cross-object contacts. While adequate for single-object compositing, this bias makes the first insertion in two-turn compositing prone to being interpreted as background, causing partial removal, distortion, over-blending, and inconsistent interactions with the subsequent compositing object.

**Parallel image-prompted compositing.** Building on these observations, we adopt a parallel strategy that simultaneously composes pairwise objects into the background, thereby preserving realistic interactions among objects and with the background. To explicitly distinguish overlap and exclusive regions, we construct the following masks from two object segments $\{\mathbf{x}_p\}_{p\in\{a,b\}}$ with binary masks $\{\mathbf{m}_p\}_{p\in\{a,b\}}$ representing the bounding boxes of the objects:

$$\mathbf{m}_u = \mathbf{m}_a \vee \mathbf{m}_b, \qquad \mathbf{m}_{ab} = \mathbf{m}_a \wedge \mathbf{m}_b, \qquad \mathbf{m}_a^{\text{ex}} = \mathbf{m}_a \wedge (1-\mathbf{m}_b), \qquad \mathbf{m}_b^{\text{ex}} = \mathbf{m}_b \wedge (1-\mathbf{m}_a). \quad (1)$$

The masked background is obtained by erasing pixels covered by the union mask,

$$\mathbf{x}_{bg} = (1 - \mathbf{m}_u) \odot \mathbf{x}. \quad (2)$$

As illustrated in Figure 3(a), our parallel compositing model $\mathcal{F}_\theta$ takes $\mathbf{x}_{bg}$ together with the objects and their masks to produce $\hat{\mathbf{x}} = \mathcal{F}_\theta(\mathbf{x}_{bg}, \{\mathbf{x}_p\}, \{\mathbf{m}_p\})$. Following latent diffusion models, each object segment $\mathbf{x}_p$ and the background $\mathbf{x}_{bg}$ are encoded into latent codes:

$$\mathbf{c}_p = E_{\text{shape}}(\mathbf{x}_p), \quad \mathbf{z}_{bg} = E_{\text{VAE}}(\mathbf{x}_{bg}), \quad p \in \{a, b\}, \tag{3}$$

which are then fused via cross-attention so that $\mathbf{z}_{bg}$ is conditioned on $\{\mathbf{c}_p\}$; the updates are spatially guided by $\mathbf{m}_a^{\text{ex}}$, $\mathbf{m}_b^{\text{ex}}$, and $\mathbf{m}_{ab}$, as detailed in Subsection 3.2. Following prior single-object compositing, we train $\mathcal{F}_\theta$ with a self-supervised recomposition objective to reconstruct image $\mathbf{x}$.

## 3.2 INTERACTION TRANSFORMER

As illustrated in Figure 3(b), each interaction transformer block applies self-attention to capture global dependencies, then employs a mask-guided Mixture-of-Experts (MoE) to route background, exclusive, and overlap regions to dedicated experts. Their outputs are gated by partition masks, merged through a residual update, and refined with a feed-forward network (FFN), ensuring spatially grounded, region-consistent updates across the image.

**Feature-space routing masks.** Object masks are originally defined in image space, while our computations are performed in feature space. We therefore downsample masks to the feature-map resolution using bilinear interpolation,

$$\overline{\mathbf{m}} = \mathcal{D}_{H,W}(\mathbf{m}), \tag{4}$$

where $H, W$ denote the spatial dimensions at each layer. From these, we obtain background masks $\overline{\mathbf{m}}_{bg} = 1 - \overline{\mathbf{m}}_u$, exclusive masks $\overline{\mathbf{m}}_a^{ex}, \overline{\mathbf{m}}_b^{ex}$, and overlap masks $\overline{\mathbf{m}}_{ab}$.

**Spatially-aware Mixture-of-Experts.** Given features $\mathbf{z}^{l-1}$ and masks, the MoE applies region-specific experts to the same input and aggregates their outputs residually to yield $\mathbf{z}^l$. Here $f_Q, f_K, f_V$ denote $1{\times}1$ projections for attention. Figure 3(c) illustrates the structure of each expert block.

*Background expert.* The background is left unchanged, i.e., $\mathbf{h}_{bg} = \mathbf{z}^{l-1}$.

*Exclusive-region experts.* For non-overlapping regions of object $p$, we inject object-specific appearance by cross-attending background queries to object codes:

$$\mathbf{h}_p = \text{CrossAttn}\Big(f_Q(\mathbf{z}^{l-1}), \ f_K(\mathbf{c}_p), \ f_V(\mathbf{c}_p)\Big), \qquad p \in \{a, b\}, \tag{5}$$

with the updates applied under the mask $\overline{\mathbf{m}}_p^{ex}$.

*Overlap expert.* In overlap regions, directly fusing two object codes with an MLP may cause blurred boundaries or inconsistent dominance. To overcome this, we introduce an attention-gated expert that adaptively favors either object, or their blend, conditioned on the background context.

We first construct a gating query from the background code:

$$\mathbf{q}_g = g_Q(\mathbf{z}^{l-1}), \tag{6}$$

where $g_Q$ is a $1{\times}1$ projection analogous to $f_Q$. This query acts as a position-wise *referee*, deciding at each spatial location whether object $a$ or $b$ should dominate.

Each object code is then aggregated into the background space via attention:

$$\tilde{\mathbf{c}}_p = \text{CrossAttn}\big(\mathbf{q}_g, \ f_K(\mathbf{c}_p), \ f_V(\mathbf{c}_p)\big), \qquad p \in \{a, b\}. \tag{7}$$

yielding $\tilde{\mathbf{c}}_p$, a per-location summary of how object $p$ aligns with the background query.

To determine the context-conditioned preference, we first score how well each aggregated object code matches the gating query and then convert the two scores into a mixing weight:

$$s_p = \frac{\langle \mathbf{q}_g, \tilde{\mathbf{c}}_p \rangle}{\sqrt{d}}, \qquad \alpha = \frac{e^{s_a/\tau}}{e^{s_a/\tau} + e^{s_b/\tau}}. \tag{8}$$

Here $d = \dim(\mathbf{q}_g)$ and $\tau > 0$ is a temperature controlling the sharpness of the selection; the gating query thus favors the object whose aggregated code best explains the local observation.

The pairwise context is then obtained by adaptive $\alpha$-blending of the aggregated object codes,

$$\mathbf{c}_{ab} \;=\; \alpha\,\tilde{\mathbf{c}}_a + (1-\alpha)\,\tilde{\mathbf{c}}_b, \tag{9}$$

providing a position-wise compatibility representation of both objects while preserving boundaries.

Finally, we inject this context into the background code through cross-attention:

$$\mathbf{h}_{ab} \;=\; \mathrm{CrossAttn}\!\big(f_Q(\mathbf{z}^{l-1}),\; f_K(\mathbf{c}_{ab}),\; f_V(\mathbf{c}_{ab})\big), \tag{10}$$

and use $\mathbf{h}_{ab}$ as the overlap expert output, yielding an order-agnostic, attention-based mechanism that adaptively selects between objects while enforcing boundary consistency.

A key property of this design is that the gating query $q_g$ carries learned occlusion semantics rather than appearance cues, as it is derived from the deep background representation $\mathbf{z}^{l-1}$. Consequently, the gating in Equation (8) performs context-guided, pairwise *arbitration* between the two objects: the softmax jointly normalizes their responses, inducing an implicit object-object interaction that determines which object should dominate at each spatial location.

**Region-gated updates and aggregation.** Expert outputs are masked by corresponding regions:

$$\Delta\mathbf{z}_{bg} = \overline{\mathbf{m}}_{bg} \odot \mathbf{h}_{bg}, \qquad \Delta\mathbf{z}_p = \overline{\mathbf{m}}_p^{ex} \odot \mathbf{h}_p, \;\; p \in \{a,b\}, \qquad \Delta\mathbf{z}_{ov} = \overline{\mathbf{m}}_{ab} \odot \mathbf{h}_{ab}. \tag{11}$$

The regional updates are then aggregated and added residually:

$$\Delta\mathbf{z} = \Delta\mathbf{z}_{bg} + \Delta\mathbf{z}_a + \Delta\mathbf{z}_b + \Delta\mathbf{z}_{ov}, \qquad \mathbf{z}^l = \mathbf{z}^{l-1} + \Delta\mathbf{z}, \tag{12}$$

after which an FFN refines $\mathbf{z}^l$ before it is passed to the next block.

## 3.3 AUGMENTATIONS

Robust compositing requires handling both *out-of-plane* viewpoint changes and *in-plane* rotations. We adopt two geometry-aware augmentations during training.

**Multi-view shape prior.** To capture viewpoint variations beyond standard 2D augmentations, we employ an off-the-shelf single-view reconstruction model to render $K$ auxiliary views. Each view is encoded by a frozen shape encoder $E_{\mathrm{shape}}$ into latent codes $\{\mathbf{c}_p^k\}_{k=1}^K$. These codes are randomly permuted, concatenated, normalized, and fused with a lightweight MLP $\mathcal{V}$:

$$\mathbf{p}_p \;=\; \mathcal{V}\big(\mathrm{LN}([\mathbf{c}_p^1;\cdots;\mathbf{c}_p^K])\big), \tag{13}$$

producing a compact multi-view descriptor that is shape-preserving.

**In-plane rotation.** To improve robustness against in-plane misalignment, we apply random rotations $\theta \sim \mathcal{U}(-\pi/6, \pi/6)$ to object images and their masks, and encode them with $E_{\mathrm{shape}}$. This enhances alignment with background context and increases robustness to in-plane transformations.

## 4 EXPERIMENTS

**Datasets.** PICS is trained on a mixture of image datasets. For validating pairwise recompositing, we use the LVIS benchmark, and for testing, we adopt DreamBooth (Ruiz et al., 2023) together with a set of in-the-wild images. Comprehensive descriptions of the datasets, as well as implementation details including network architecture, training and inference settings, are provided in the appendix.

**Evaluation metrics.** We evaluate recompositing quality both on the entire images and on bounding box intersection regions using PSNR, SSIM, and LPIPS. To further assess the realism of the generated images, we employ CLIP-Score (Hessel et al., 2021), DINOv2-Score (Oquab et al., 2024), and DreamSim (Fu et al., 2023). For the image compositing task, we specifically adopt CLIP-Score, DINOv2-Score, and DreamSim for evaluating the compositing quality.

Table 1: Quantitative comparison of object recompositing against prior methods on the LVIS validation set. The prefix "m-" indicates evaluation restricted to the intersection regions. **Bold** numbers denote the best performance, and underlined numbers indicate the second best.

| Method | mPSNR ↑ | mSSIM ↑ | mLPIPS ↓ | PSNR ↑ | FID ↓ | LPIPS ↓ | CLIP-score ↑ | DINOV2-score ↑ | DreamSim ↓ |
|---|---|---|---|---|---|---|---|---|---|
| PbE (CVPR'23) | 10.24 | 0.4241 | 0.4535 | 15.29 | 34.93 | 0.4138 | 81.42 | 0.4320 | 0.4896 |
| ControlCom (arXiv'23) | 11.82 | 0.3185 | 0.3986 | 17.61 | 26.93 | 0.3375 | **85.39** | 0.5264 | 0.3248 |
| ObjectStitch (CVPR'23) | 10.84 | 0.3471 | 0.4203 | 16.55 | 29.68 | 0.3572 | 85.01 | 0.5574 | 0.3458 |
| AnyDoor (CVPR'24) | 11.62 | 0.5283 | 0.4185 | 17.12 | 27.17 | 0.3302 | 84.99 | **0.6089** | 0.2820 |
| OmniPaint (ICCV'25) | 12.20 | 0.3096 | 0.4618 | 16.09 | 26.25 | 0.3542 | 83.11 | 0.5673 | 0.2774 |
| PICS (ours) | **13.88** | **0.5823** | **0.3221** | **18.27** | **24.99** | **0.2530** | 85.25 | 0.5713 | **0.2659** |

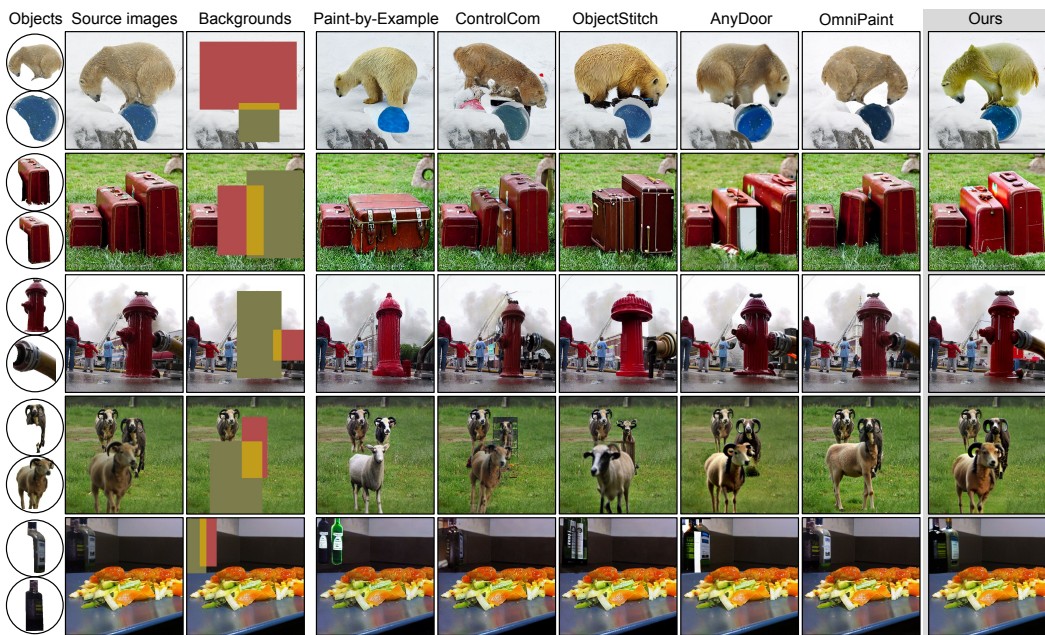

Figure 4: Qualitative comparison on the LVIS validation set. Source images, backgrounds, and the two decomposed objects are shown on the left. On the right are the recompositing results from different methods. Our approach is the only one that produces composites with realistic spatial interactions between scene objects while maintaining scene consistency and object identity.

## 4.1 OBJECT RECOMPOSITING

**Qualitative comparison.** Object recompositing refers to compositing objects and backgrounds from the same source image, which serves as our evaluation setting. We compare our method with five prior approaches, namely PbE (Yang et al., 2023), ControlCom (Zhang et al., 2023), ObjectStitch (Song et al., 2023), AnyDoor (Chen et al., 2024b) and OmniPaint (Yu et al., 2025). The baselines adopt a two-step compositing protocol, where the red region is placed first and the green region second, whereas our method performs parallel pairwise compositing. As illustrated in Figure 4, existing methods primarily designed for single-object compositing struggle to generate clear features in occluded regions. While the objects appear harmonized with the background, these methods often fail to handle occlusion order correctly and may introduce artifacts by improperly layering one object over another. In contrast, PICS consistently generates recompositions that preserve object identity while maintaining coherent and plausible connectivity across interacting regions.

**Quantitative comparison.** As reported in Table 1, our method delivers consistent improvements over competing approaches in PSNR, SSIM, FID, and LPIPS, including evaluations on intersection regions, demonstrating its ability to faithfully capture the data distribution. While AnyDoor achieves slightly higher DINO-v2 scores, this advantage is partly attributable to its use of additional edge maps as input, which aids semantic preservation but limits flexibility when the structural alignment between the input and background is poor, often resulting in inferior scene consistency.

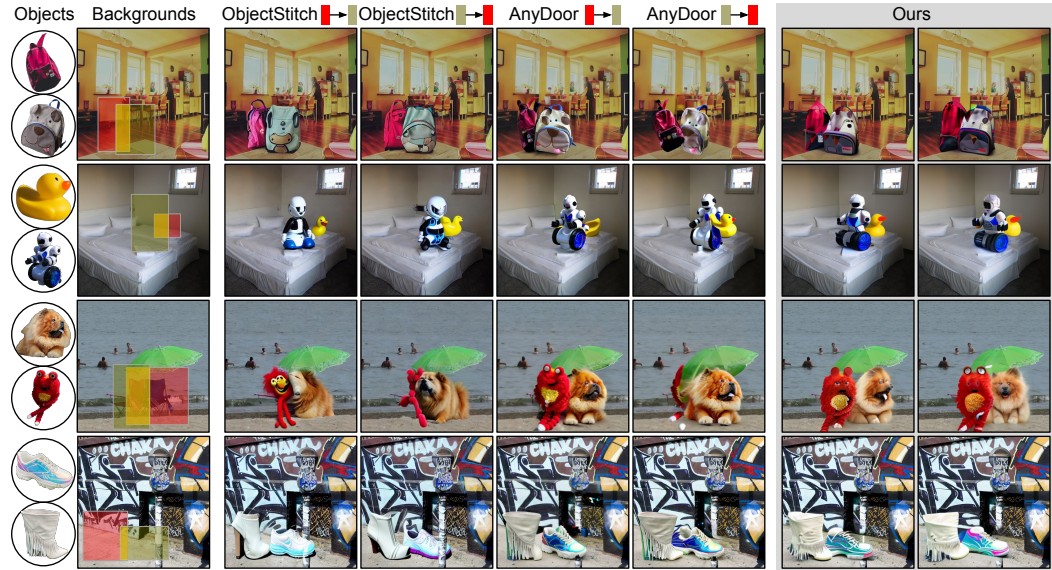

Figure 5: Qualitative comparison of different composition orders on the DreamBooth test set. Left: backgrounds and two objects. Right: results from different methods. Our approach better preserves natural contacts and occlusions, while implicitly learning the correct occlusion order.

Table 2: Quantitative comparison of pairwise object compositing on the DreamBooth testing set. **Bold** numbers denote the best performance, and underlined numbers indicate the second best.

| Method | FID ↓ | CLIP-score ↑ | DINOV2-score ↑ | DreamSim ↓ |
|---|---|---|---|---|
| PbE (CVPR'23) | 262.4 | 51.95 | 0.2383 | 0.4321 |
| ControlCom (arXiv'23) | 273.4 | 52.38 | 0.2414 | 0.3194 |
| ObjectStitch (CVPR'23) | 260.4 | 51.35 | 0.3203 | 0.3374 |
| AnyDoor (CVPR'24) | 274.1 | 51.24 | 0.3401 | 0.2733 |
| FreeCompose (ECCV'24) | 299.6 | 51.71 | 0.2157 | 0.3521 |
| OmniPaint (ICCV'25) | 260.4 | 50.32 | **0.3741** | **0.2632** |
| InsertAnything (AAAI'26) | 266.0 | 50.54 | 0.3612 | 0.2934 |
| PICS (ours) | **255.5** | **54.02** | 0.3631 | 0.3054 |

Table 3: User study (%). "Quality", "Fidelity", and "Consistency" evaluate image realism, identity preservation, and object coherence, respectively.

| Method | Quality ↑ | Fidelity ↑ | Consistency ↑ |
|---|---|---|---|
| PbE | 5.13 | 2.53 | 8.70 |
| ControlCom | 12.2 | 15.2 | 13.0 |
| ObjectStitch | 12.8 | 7.59 | 15.9 |
| AnyDoor | 14.1 | 18.4 | 12.3 |
| FreeCompose | 2.56 | 1.27 | 4.35 |
| OmniPaint | 17.3 | **19.0** | 10.9 |
| InsertAnything | 16.0 | 18.4 | 12.3 |
| PICS (ours) | **17.7** | 17.7 | **22.5** |

## 4.2 OBJECT COMPOSITING

**Qualitative comparison.** We compare our pairwise object compositing results with ObjectStitch and AnyDoor, using their default settings and pretrained models to sequentially compose objects into backgrounds using the DreamBooth testing set. As shown in Figure 5, both ObjectStitch and AnyDoor exhibit boundary artifacts when a newly inserted object partially occludes a previously composed one. AnyDoor often causes the current object to either completely cover, entangle with, or shrink parts of the previous composed object, while ObjectStitch struggles to preserve object identity. In comparison, our method produces more boundary-consistent compositions.

**Quantitative comparison.** Table 2 reports the quantitative comparisons across various evaluation metrics. Our method achieves the best overall performance. On DreamSim, both of AnyDoor and OmniPaint attain higher scores, where AnyDoor leverages high-frequency object features as additional guidance, which helps preserve object structure but at the cost of consistent compositing with the background, and OmniPaint, on the other hand, is built upon a flow-matching FLUX backbone whose generative prior is stronger than standard diffusion.

**User study.** We conducted a user study with 20 participants to evaluate object compositing quality, focusing on realism, object fidelity, and intersection quality. Using the same objects and backgrounds from the challenging DreamBooth dataset, participants were asked to rank and score results from different methods. As shown in Table 3, our approach outperforms prior methods in terms

Table 4: Ablation study on in-the-wild test set to verify key components of our method. "ITB" denotes the interaction transformer block, "SV" single-view, "Rot." the rotation augmentation, "MV" the multi-view augmentation, and "Comb." the combined data.

| No. | MLP | ITB | SV | Rot. | MV | LVIS | Comb. | FID $\downarrow$ | CLIP-score $\uparrow$ |
|---|---|---|---|---|---|---|---|---|---|
| 1 | ✓ | | ✓ | | | | ✓ | 173.1 | 74.6 |
| 2 | | ✓ | ✓ | | | | ✓ | 165.2 | 76.3 |
| 3 | | ✓ | ✓ | ✓ | | | ✓ | 162.5 | 74.9 |
| 4 | | ✓ | ✓ | ✓ | | ✓ | | 158.2 | 77.3 |
| 5 | | ✓ | ✓ | ✓ | ✓ | | ✓ | **151.3** | **79.1** |

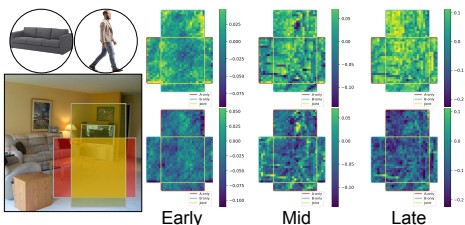

Figure 6: $\Delta s$ across denoising stages. Top: $a \rightarrow$ sofa, $b \rightarrow$ human; bottom: swapped.

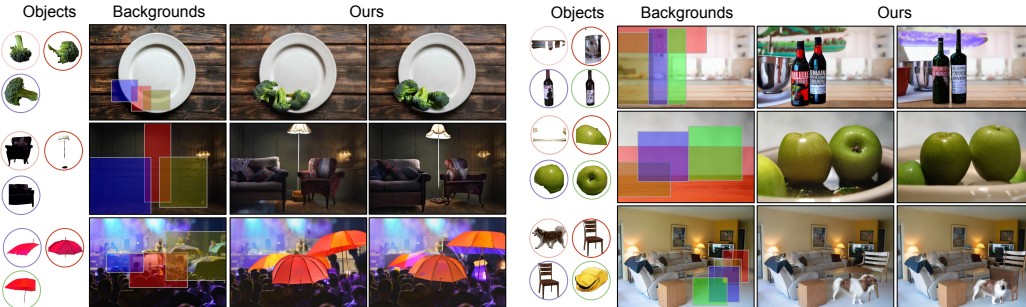

Figure 7: 3-object compositing.

Figure 8: 4-object compositing.

of realism and consistency, underscoring its effectiveness. On the fidelity criterion, our method performs comparably to AnyDoor, as both leverage the DINOv2 model to encode identity features.

## 4.3 ABLATION STUDIES

**Module choices.** Table 4 summarizes ablations on the in-the-wild test set over three factors: model architecture, geometry-aware augmentation, and training data scale. In the MLP baseline, the two object codes are concatenated and passed through an MLP to model their interaction, and the resulting representation is then fused with the background via cross-attention. Moving from Setting 1 to Setting 2, replacing the MLP with the proposed interaction transformer block for intersection modeling consistently improves all metrics, reflecting stronger reasoning over inter-object cues in overlapping regions. Introducing geometry-aware augmentations further enhances robustness: in-plane rotation (Setting 3) mitigates misalignment within the image plane, while the multi-view prior (Setting 4) improves robustness to viewpoint variation. Expanding the training set from LVIS-only to the full 1M-image collection (Setting 5) provides the most significant gain, improving generalization to unseen object-background pairings.

$\alpha$-**blending.** We evaluate whether the overlap expert learns spatially resolved mixing weights in the multi-scale feature spaces. The coefficient $\alpha$ is derived from the score difference $\Delta s = s_a - s_b$, where positive values favor object $a$, negative values favor object $b$, and values close to zero lead to a balanced blend, with $\alpha = \sigma(\Delta s / \tau)$ (Equation (8)) ensuring that $\Delta s$ and $\alpha$ vary consistently. Figure 6 shows $\Delta s$ under two indexing choices: in the top row, $a$ corresponds to the sofa and $b$ to the human, while in the bottom row the roles are swapped but their compositing regions remain unchanged. The sign of $\Delta s$ is consistently aligned with visibility, being positive where the human remains visible and negative elsewhere, and it reverses accordingly when the indices are exchanged. This confirms that the gating mechanism encodes actual visibility relationships rather than relying on the arbitrary order of inputs, thereby realizing the intended logistic blending at each spatial location. Furthermore, across denoising steps during inference (early, mid, late), the maps evolve progressively: they begin coarse and noisy, become spatially decisive mid-way, and ultimately sharpen into fine-grained boundaries, reflecting refinement dynamics characteristic of diffusion models.

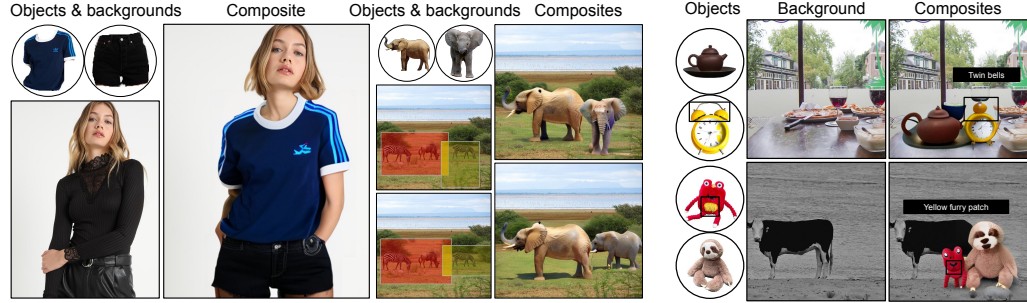

Figure 9: Applications. Virtual try-on; novel-view compositing.

Figure 10: Failure cases.

## 4.4 GENERALIZATION TO MULTI-OBJECT COMPOSITING

To assess the scalability of our approach beyond the pairwise setting, we additionally train two models for 3-object and 4-object compositing using samples constructed entirely from the LVIS dataset. Representative results are shown in Figures 7 and 8. In the 3-object setting, the composite results reflect consistent occlusion ordering and contact relations, and object identities remain well preserved even where multiple masks intersect, indicating that our interaction module distributes appearance features without collapsing fine details. The 4-object setting presents more entangled configurations, including multi-level occlusions. The model remains stable: as shown in the bottom example of Figure 8, the backpack is almost fully occluded and is correctly omitted in the final composite, indicating that the model respects visibility rather than hallucinating hidden content.

## 4.5 APPLICATIONS

**Virtual try-on.** PICS supports pairwise try-on of an upper-body garment and a lower-body garment. As shown in Figure 9 (left), it maintains a clean, well-aligned seam between the two garments and handles overlap robustly, avoiding color bleeding and double edges even under moderate pose changes. Additional side-by-side comparisons with recent methods are provided in appendix K.2.

**Novel-view composition.** Our approach also supports novel-view composition, which generates a previously unseen viewpoint of an object and harmonizing it with the background to ensure visual coherence; see Figure 9 (right). For example, when the bounding box is horizontally elongated, the model correctly generates a side view of the elephant. This demonstrates the ability of our framework to capture spatial priors and to contextually compose objects.

## 5 CONCLUSION AND OUTLOOK

We presented PICS, a parallel paradigm for pairwise image compositing that explicitly models spatial interactions among objects and the background. Central to the method is an Interaction Transformer with mask-guided experts and an adaptive $\alpha$-blending mechanism that enables region-aware composition with boundary fidelity. Robustness to geometric variation is further improved by geometry-aware augmentations that address both out-of-plane and in-plane pose changes.

While PICS yields high-quality pairwise composites, it naturally extends to multi-object scenarios via parallel compositing, enabling sophisticated composition in complex scenes. In addition, as shown in Figure 10, we observe occasional geometry and texture degradation in extremely cluttered environments, attributable to the limited capacity of the shape encoder. Future work will further optimize the routing and fusion mechanisms to enhance the scalability of multi-object compositing while guaranteeing semantic fidelity under diverse lighting conditions.

**Ethics statement.** This work is conducted on publicly available datasets and is intended solely for scientific research.

**Reproducibility statement.** We have made our best effort to ensure reproducibility, including but not limited to: 1) dataset description in appendix A and implementation details in appendix B; 2) detailed graphic illustrations of model architectures, training and inference in Figures 3, 14, 13; and 3) source code and checkpoints.

**Acknowledgments.** This work was partly supported by NSERC Discovery, CFI-JELF, NSERC Alliance, Alberta Innovates and PrairiesCan grants.

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

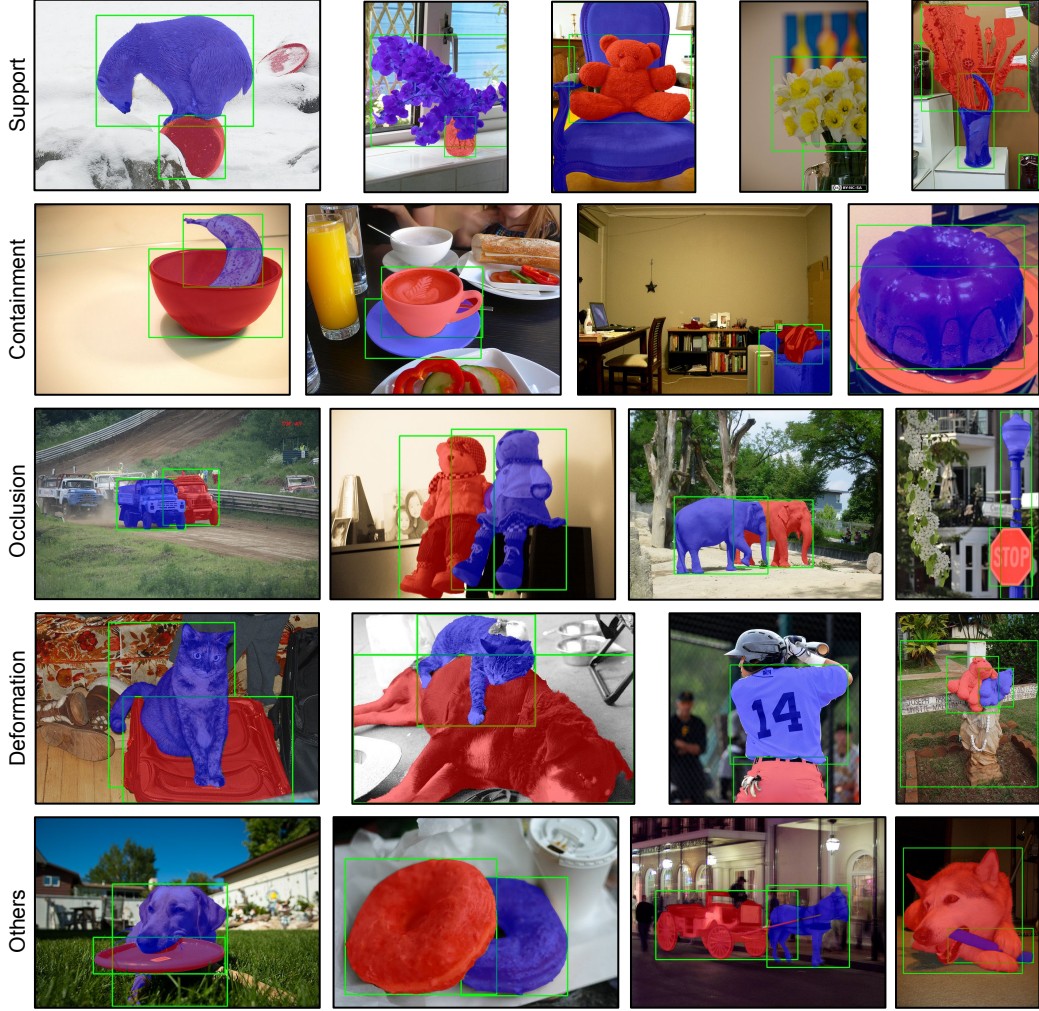

Figure 11: Examples of pairwise object relations from the LVIS validation set, with object instances visualized alongside their bounding boxes.

# A    DATASET

## A.1    PAIRWISE OBJECTS VISUALIZATION

As discussed in the main paper, pairwise object interactions are ubiquitous and pervasive across diverse real-world datasets. Here, we present several illustrative examples in Figure 11. For each type of interaction, including support, containment, occlusion, deformation, and an additional category that encompasses less canonical or atypical cases, we provide a few representative examples with object instances visualized alongside their bounding boxes.

Specifically, support relations include cases such as a bear standing on a ball, a toy placed on a chair, and a bouquet supported by a vase; containment is exemplified by donuts arranged in a box, fruit in a bowl, and flowers inside a vase; occlusion is demonstrated through an elephant obscured by another elephant, and a vehicle blocked by another vehicle; deformation is shown by wrinkles formed between overlapping clothes and pants, toys compressed against each other, and a soft bag deformed under weight. The others category includes diverse interactions such as a dog holding a plate in its mouth, objects leaning against each other, or items stacked closely together, which are not easily assigned to the primary relation types. These examples highlight that modeling pair-

Table 5: Statistics and description of our training datasets for pairwise object compositing.

| Datasets | #Training | Type |
|---|---|---|
| LVIS (Gupta et al., 2019) | 34,160 | General |
| VITON-HD (Choi et al., 2021) | 11,647 | Try-on |
| Objects365 (Shao et al., 2019) | 940,764 | General |
| Cityscapes (Cordts et al., 2016) | 536 | Street |
| Mapillary Vistas (Neuhold et al., 2017) | 603 | Street |
| BDD100K (Yu et al., 2020) | 1,012 | Street |

wise object interactions is essential for realistic scene compositing, ensuring consistent boundaries, plausible occlusion, and physically coherent interactions.

## A.2 PAIRWISE SPATIAL RELATION

To analyze the spatial relationships between pairwise bounding boxes, we construct a single aggregated heatmap over their overlapping regions, as visualized in Figure 12. For each pair of intersecting boxes randomly sampled from the LVIS training set (10k samples), we normalize one bounding box to the canonical $[0, 1] \times [0, 1]$ heatmap coordinate frame, and project the other box accordingly. The overlapping region directly contributes to the heatmap values, and aggregating over all samples produces the final distribution of overlap regions. The heatmap peaks near the image center $(0.5, 0.5)$, clearly indicating that a majority of bounding box pairs exhibit significant overlap in this region (approximately $90\%$). Even in peripheral regions with minimal overlap, nearly $50\%$ of bounding boxes still intersect, further highlighting the prevalence of spatial interactions across the training data.

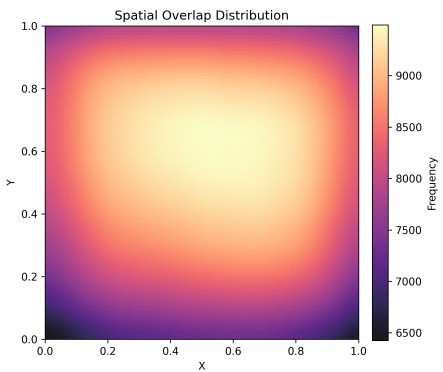

Figure 12: Heatmap of spatial relations of pairwise bounding boxes.

## A.3 TRAINING DATASET PREPARATION

To effectively train our model for pairwise object compositing, we curated a large-scale dataset consisting of nearly 1 million diverse samples by combining real-world datasets originally designed for object-centric scene understanding and visual try-on; see Table 5. Since not every image naturally contains multiple bounding boxes with intersections, we filtered the dataset to retain only samples with at least two intersecting boxes. Furthermore, to better facilitate effective modeling of overlapping regions, we preferentially select bounding box pairs with the highest IoU. The detailed implementation procedure is elaborated in Algorithm 1. For datasets such as Objects365 (Shao et al., 2019), which do not provide segmentation masks, we use the existing bounding box annotations as prompts for SAM (Kirillov et al., 2023) to obtain sufficiently accurate object masks. For TF-ICON (Lu et al., 2023), we discard bounding boxes corresponding to background segments.

## A.4 TESTING DATASET

We curate a 110-case test set: 80 cases use backgrounds from the LVIS validation set with object exemplars from DreamBooth, and 30 cases use Internet backgrounds with in-the-wild object images. Target insertion regions are manually annotated.

## B IMPLEMENTATION DETAILS

Our models are implemented using PyTorch, with details of the model architecture, training, and inference provided below.

**Algorithm 1** Pseudocode of pairwise bounding boxes selection algorithm in a PyTorch-like style.

```
"""
Select a pair of bounding boxes with the highest IoU
"""
def select_boxes(bbox_xyxy):
    """
    bbox_xyxy: list of bounding boxes in [x0, y0, x1, y1] format
    return: tuple of two bounding boxes with highest IoU, or -1 if none exist
    """
    if len(bbox_xyxy) <= 2:
        return -1

    # Compute pairwise IoU matrix
    iou_matrix = compute_iou_matrix(bbox_xyxy) # assume this function exists

    # Find the pair with maximum IoU
    index0, index1 = np.unravel_index(np.argmax(iou_matrix), iou_matrix.shape)
    max_iou = iou_matrix[index0, index1]

    if max_iou <= 0:
        return -1

    return bbox_xyxy[index0], bbox_xyxy[index1]
```

## B.1 MODEL ARCHITECTURE

Our framework builds upon the publicly available implementations of Stable Diffusion v2.1 and ControlNet v1.0. In particular, we adopt the U-Net backbone of Stable Diffusion as the generative model and augment it with a ControlNet branch to enable spatially guided compositing. The control scale is fixed to $1.0$ throughout training and inference to ensure a balanced contribution from both the generative and control pathways. To enhance interaction modeling, we systematically replace the original residual blocks with our proposed Interaction Transformer blocks. Specifically, all 25 blocks of the U-Net including 12 encoder blocks, 1 middle block, and 12 decoder blocks are substituted with IT blocks. In parallel, the ControlNet branch also undergoes the same replacement, where all 13 blocks (12 encoder and 1 middle block) are re-implemented using our IT design. This consistent replacement ensures that both the generative and control pathways benefit from the improved modeling capacity of IT blocks, thereby strengthening cross-object reasoning and compositing fidelity. Additionally, similar to the original ControlNet, the connections from the control model to the generation model are initialized by zero-convolutions, which prevents the generative capabilities of the controlled U-Net from diminishing at the beginning of training.

As shown in Figure 13, for the multi-view shape encoder, we render six novel views of each object using the single-view reconstruction model Zero123++ (Shi et al., 2023). Each view is encoded into a latent representation by a pretrained DINOv2 image encoder (Oquab et al., 2024), which provides strong semantic and structural features. The resulting per-view codes are aggregated by our fusion module to form a compact multi-view descriptor, enriching the object representation with both global shape priors and fine-grained texture details. The MLP used for the multi-view shape prior is implemented as two-layer feedforward networks.

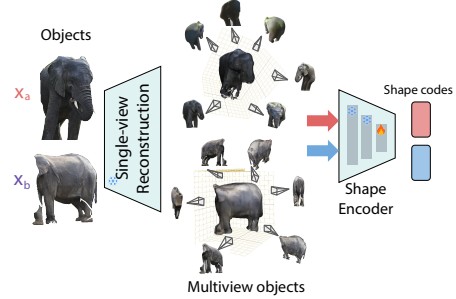

Figure 13: Details of multi-view shape prior.

## B.2 TRAINING DETAILS

For the objective loss function, we adopt the standard denoising diffusion objective, defined as the mean squared error between the predicted noise and the ground-truth Gaussian noise:

$$\mathcal{L}(\theta) = \mathbb{E}_{\mathbf{x}_{bg}, \{\mathbf{x}_p, \mathbf{m}_p\}, t, \varepsilon \sim \mathcal{N}(0,1)} \left[ \left\| \mathcal{F}_\theta \big( \mathbf{x}_{bg}, \{\mathbf{x}_p, \mathbf{m}_p\}_{p \in \{a,b\}}, t \big) - \varepsilon \right\|_2^2 \right]. \quad (14)$$

The temperature parameter is set to $\tau = 0.5$ for all experiments. Our model is implemented in PyTorch Lightning and trained with mixed-precision (fp16) on NVIDIA H100 GPUs with 80GB

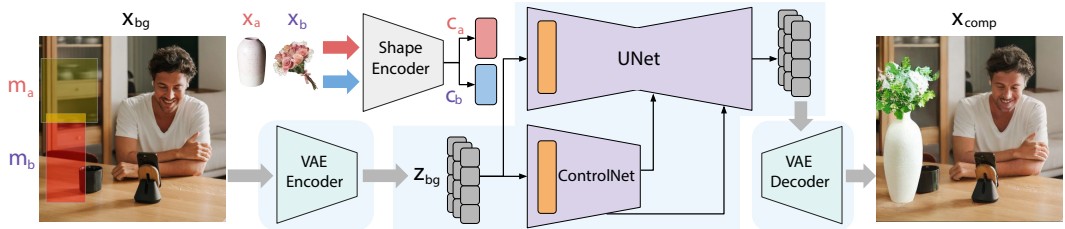

Figure 14: Inference process of PICS. The background and object embeddings, together with their masks, are fused in latent space and decoded by the VAE to produce the final composite $\mathbf{x}_{comp}$.

memory. We train with a batch size of 8, using Adam (Kingma, 2014) with a learning rate of $1 \times 10^{-5}$ and gradient accumulation of 1.

### B.3 INFERENCE DETAILS

As illustrated in Figure 14, during inference our model takes the background embedding $\mathbf{z}_{bg}$, obtained by encoding the background image $\mathbf{x}_{bg}$ with the VAE encoder, and the object codes $\mathbf{c}_a$ and $\mathbf{c}_b$, extracted from the object images $\mathbf{x}_a$ and $\mathbf{x}_b$ using shape encoders, together with their corresponding compositing masks $\mathbf{m}_a$ and $\mathbf{m}_b$. These components are fused to form a latent composite, which is subsequently decoded by the VAE decoder to generate the final composite image $\mathbf{x}_{comp}$. Specifically, the DDIM sampler generates the composite image after 50 denoising steps, with a classifier-free guidance scale of 5.0 (Ho & Salimans, 2021).

## C MULTI-OBJECT COMPOSITING

### C.1 MATHEMATICAL FORMULATION

We extend the pairwise overlap expert to the case of $M$ composed objects.

*Multi-object overlap expert.* Given object codes $\{\mathbf{c}_1, \ldots, \mathbf{c}_M\}$ and the background feature $\mathbf{z}^{l-1}$, the gating query is computed as

$$\mathbf{q}_g = g_Q(\mathbf{z}^{l-1}). \tag{15}$$

Each object code is aligned to the background query via cross-attention:

$$\tilde{\mathbf{c}}_p = \text{CrossAttn}\big(\mathbf{q}_g, \ f_K(\mathbf{c}_p), \ f_V(\mathbf{c}_p)\big), \qquad p = 1, \ldots, M. \tag{16}$$

A compatibility score is computed for every aggregated object:

$$s_p = \frac{\langle \mathbf{q}_g, \tilde{\mathbf{c}}_p \rangle}{\sqrt{d}}, \qquad p = 1, \ldots, M, \tag{17}$$

and normalized via a softmax:

$$\alpha_p = \frac{\exp(s_p/\tau)}{\sum_{j=1}^{M} \exp(s_j/\tau)}, \qquad p = 1, \ldots, M. \tag{18}$$

The fused multi-object context is obtained by an attention-weighted combination:

$$\mathbf{c}_{1:M} = \sum_{p=1}^{M} \alpha_p \, \tilde{\mathbf{c}}_p. \tag{19}$$

We then calculate a unified overlap response using background-guided cross-attention:

$$\mathbf{h}_{1:M} = \text{CrossAttn}\big(f_Q(\mathbf{z}^{l-1}), \ f_K(\mathbf{c}_{1:M}), \ f_V(\mathbf{c}_{1:M})\big). \tag{20}$$

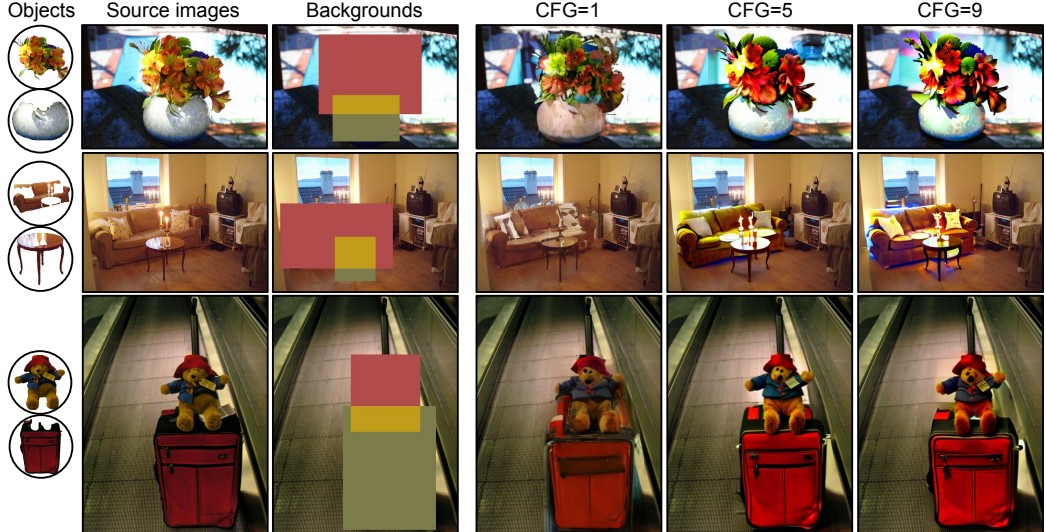

Figure 15: Effect of different classifier-free guidance (CFG) scales on image recompositing, comparing CFG values of 1, 5, and 9.

This produces an order-agnostic, attention-based overlap expert that synthesizes multi-way interaction patterns among all objects.

*Region-gated updates and aggregation.* For $M$ objects, each expert output is masked by its corresponding spatial region:

$$\Delta \mathbf{z}_{bg} = \overline{\mathbf{m}}_{bg} \odot \mathbf{h}_{bg}, \qquad \Delta \mathbf{z}_p = \overline{\mathbf{m}}_p^{ex} \odot \mathbf{h}_p, \;\; p = 1, \ldots, M, \qquad \Delta \mathbf{z}_{1:M} = \overline{\mathbf{m}}_{1:M} \odot \mathbf{h}_{1:M}. \quad (21)$$

All regional updates are aggregated via a residual update:

$$\Delta \mathbf{z} = \Delta \mathbf{z}_{bg} + \sum_{p=1}^{M} \Delta \mathbf{z}_p + \Delta \mathbf{z}_{1:M}, \qquad \mathbf{z}^l = \mathbf{z}^{l-1} + \Delta \mathbf{z}, \quad (22)$$

after which a feed-forward network refines $\mathbf{z}^l$ before passing it to the next block. This formulation reduces to the two-object case in the main text when $M = 2$.

## C.2    DATASET PREPARATION

For each image, we first discard very small instances and keep only objects above an area threshold. We then convert all remaining bounding boxes into a consistent format and compute the pairwise IoU among them. To select a set of overlapping objects, we identify the anchor, defined as the object that overlaps the most with the others. We then take the anchor's top overlapping neighbors (those with positive IoU) and choose as many as needed for the target setting. For example, the top two neighbors for a 3-object sample, or the top three neighbors for a 4-object sample. This ensures that all selected objects overlap with the anchor, although they are not required to overlap with one another. For each selected object, we extract its binary mask and the corresponding cropped RGB patch. These masks and patches are saved together with the original image to form a structured multi-object sample.

## D    EFFECT OF CLASSIFIER-FREE GUIDANCE

In our experiments, we systematically evaluate the effect of the classifier-free guidance (CFG) scale on image compositing quality by comparing three representative values: 1, 5, and 9. As shown in Figure 15, when the CFG is set to 1, the model behaves almost unconditionally, leading to noisy and blurry synthesis in the compositing regions. Interestingly, although the fidelity of the inserted objects

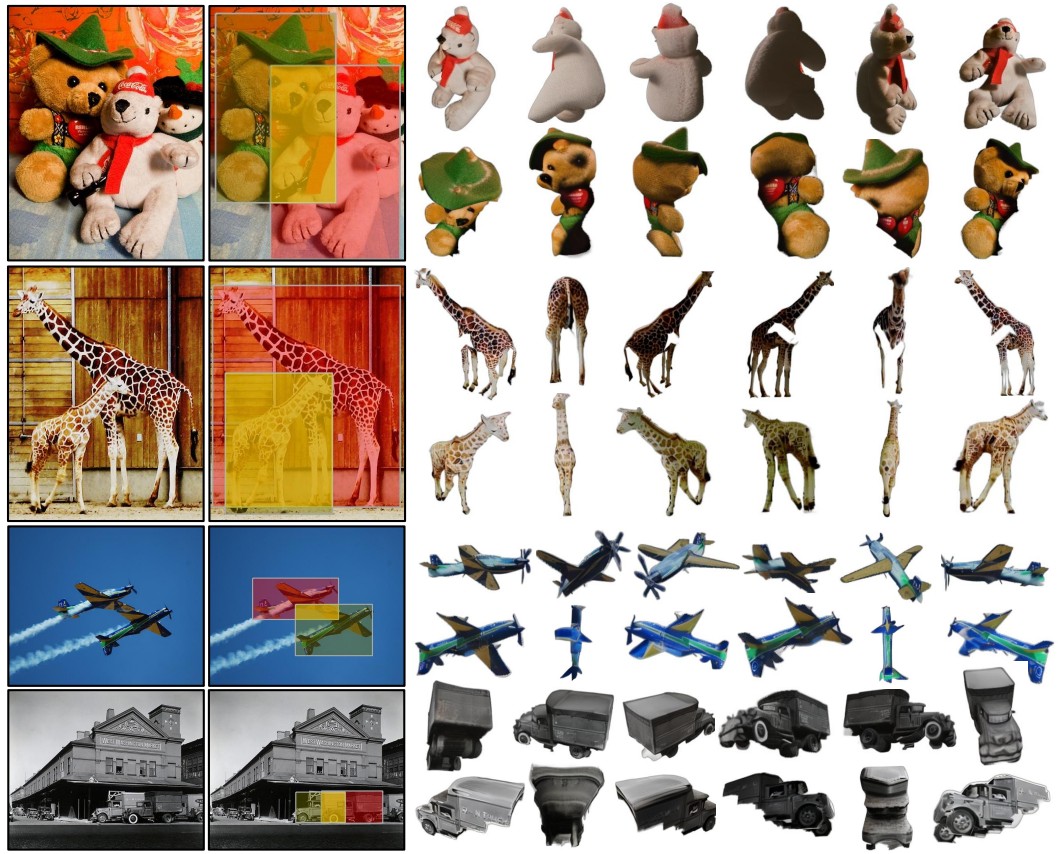

Figure 16: Four exemplars for (non-)partial 3D reconstruction. From left to right: source image, image with object bounding boxes, 6-view reconstructed object images.

is poor, the color distribution tends to match the background more naturally, resulting in better chromatic consistency. In contrast, a large CFG value such as 9 enforces strong adherence to the object condition, thereby producing composites that better preserve the identity and fine details of the reference objects. However, this often comes at the cost of visual harmony, as the object colors may deviate noticeably from the background, leading to less coherent composition. A mid-range CFG of 5 provides a favorable balance between these two extremes, ensuring that object identity is retained while maintaining reasonable consistency with the background. This observation is consistent with prior findings in guided diffusion, where overly low scales reduce conditional fidelity and overly high scales overfit the conditioning signal, thereby compromising overall realism. Hence, we adopt 5 as the default setting in all our experiments.

## E    3D RECONSTRUCTION

As shown in Figure 16, we evaluate a pretrained 3D reconstruction model, Zero123++ on occluded 2D segments from LVIS and find that, despite occlusion, it reliably reconstructs coherent *partial* 3D shapes. The partial 3D shape formed from such multi-view images provides compact features of the objects that guide the modeling of intersection regions, leading to geometrically consistent interactions and improving compositing quality.

## F    CHOICES OF OBJECT MASK

We further assess the robustness of our method to segmentation masks of varying quality, as illustrated in Figure 17. Specifically, we compare pairwise compositing results using coarse masks

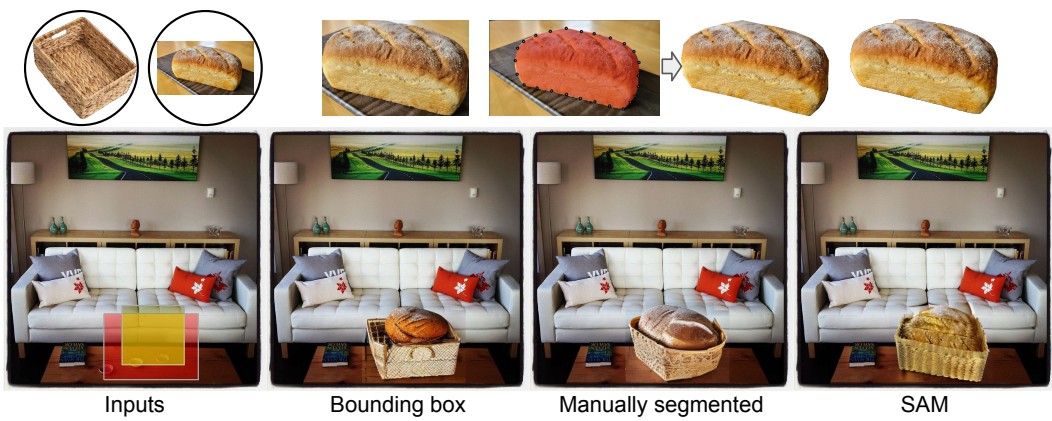

Figure 17: Four exemplars for (non-)partial 3D reconstruction. From left to right: source image, image with object bounding boxes, 6-view reconstructed object images.

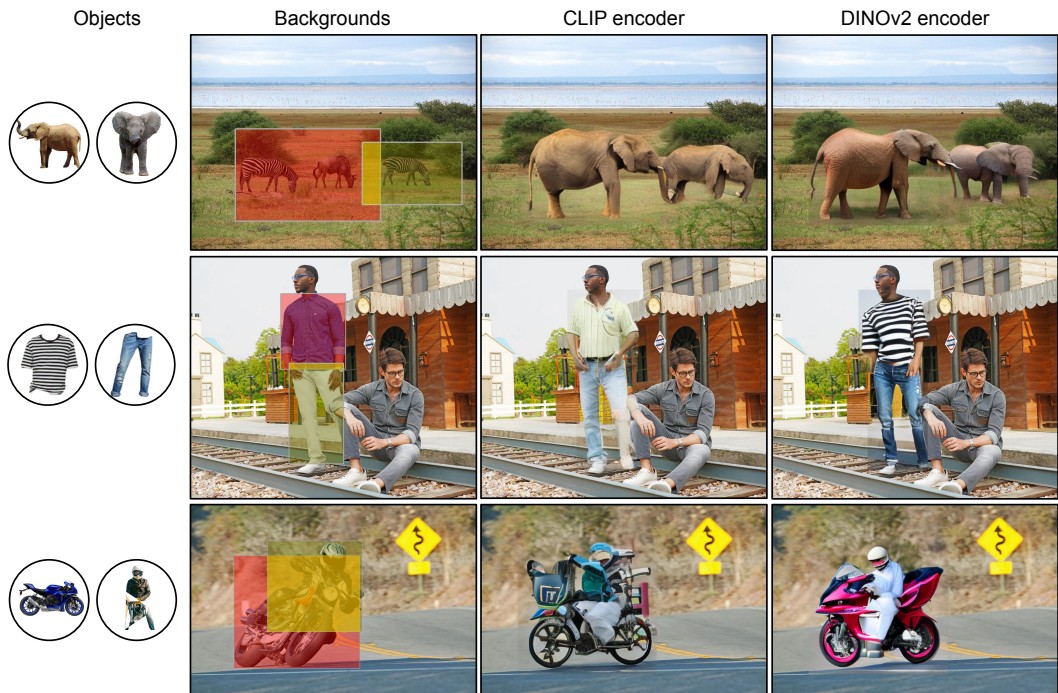

Figure 18: CLIP encoder *vs* DINOv2 encoder. Models are separately trained on LVIS dataset.

including bounding boxes and manually annotated masks[2] against results obtained with high-quality SAM masks. Our findings show a clear trend: better segmentation masks lead to better compositing. In contrast, coarse or inaccurate masks tend to introduce undesired background cues from the object image, ultimately degrading the compositing quality.

## G  CHOICES OF OBJECT ENCODER

We conduct an ablation study to evaluate how the choice of object encoder influences the fidelity of composed objects, as exemplified in Figure 18. Specifically, we replace our default object encoder with CLIP, while keeping all other components unchanged, and both of the two models are trained

---

[2]https://pixlab.io/annotate

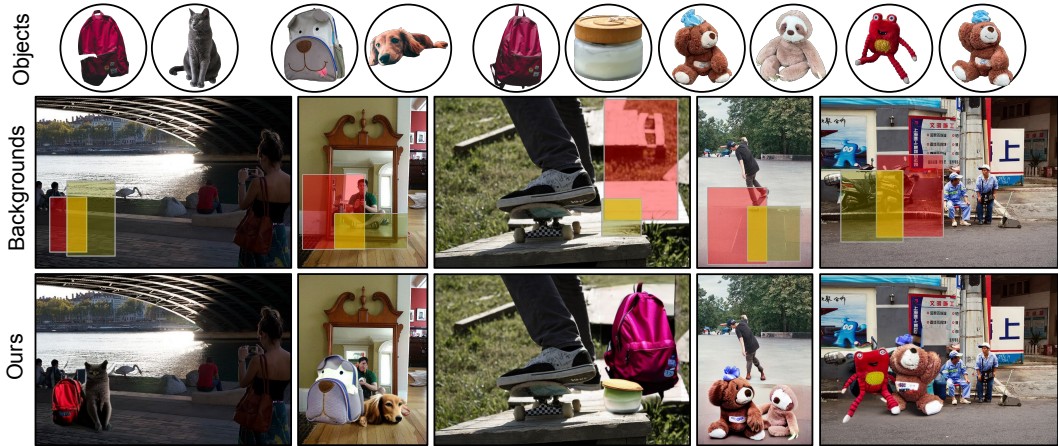

Figure 19: Additional visualization of our pairwise compositing with unoccluded inputs.

only on the LVIS dataset. The comparison reveals a clear degradation in appearance preservation when using CLIP: the composed objects exhibit diminished fine-grained details, such as the subtle skin-tone differences between the two elephants, the striped patterns on the T-shirt, and the features of the motorcyclist. These observations highlight the importance of using a stronger encoder for capturing high-frequency object textures that are crucial for identity-preserving compositing.

## H VISUALIZATION OF UNOCCLUDED INPUTS

Our model is trained exclusively on occluded object instances, without ever seeing intact objects during training. To evaluate robustness beyond this training regime, we conduct additional evaluation experiments using fully visible (unoccluded) objects as inputs, as demonstrated in Figure 19. Our method demonstrates strong generalization to this setting, where it is able to accurately generate pairwise spatial relations while preserving the identity of composed objects.

## I VISUALIZATION OF 3D AUGMENTATION

In Figure 20, we compare results generated with and without the proposed 3D augmentation strategy. Incorporating 3D augmentation enables the model to synthesize a broader range of viewpoint variations for the composed objects. For instance, in the first row, the inserted donut exhibits clear geometric changes; in the second row, the flower is rendered from a novel orientation relative to its input view; and in the last row, the doll also appears under a noticeably different facing direction. These examples illustrate that coupling our model with 3D augmentation substantially improves viewpoint diversity, leading to richer and more flexible compositional generations.

## J COMPUTATION AND TIME COST

Our model exhibits competitive computational efficiency relative to existing methods, requiring approximately 20 seconds per sample for pairwise image compositing, as shown in Table 6. While integrating the plug-and-play 3D reconstruction module introduces additional computational overhead, the overall inference-time cost remains practical.

## K MORE QUALITATIVE RESULTS

### K.1 IN-THE-WILD PAIRWISE COMPOSITING

We provide comparison results on the in-the-wild images in Figure 21. Typical artifacts observed for each sample are listed in Table 7.

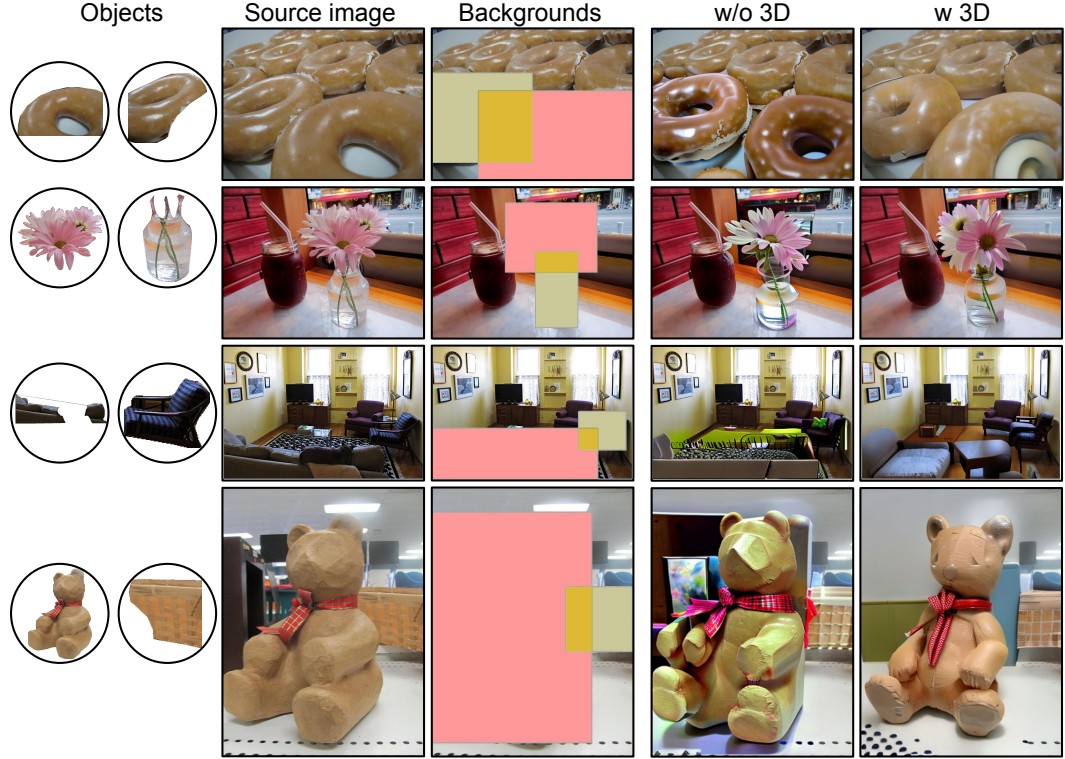

Figure 20: Additional visualization of w *vs* w/o 3D augmentation of our method.

Table 6: Comparison of inference-time computation cost for different methods.

| Methods | Params | Time (min) | GFLOPs | GPU Memory |
|---|---|---|---|---|
| PbE (Yang et al., 2023) | 1.31 G | 0.722 | 1048 | 8.70 G |
| ControlCom (Zhang et al., 2023) | 1.37 G | 0.684 | 1123 | 21.5 G |
| ObjectStitch (Song et al., 2023) | 1.31 G | 0.241 | 852 | 7.04 G |
| AnyDoor (Chen et al., 2024b) | 2.45 G | 0.342 | 2450 | 15.8 G |
| FreeCompose (Chen et al., 2024c) | 1.07 G | 1.912 | 678 | 5.79 G |
| OmniPaint (Yu et al., 2025) | 12.1 G | 1.516 | 33035 | 23.1 G |
| InsertAnything (Song et al., 2026) | 0.52 G | 1.379 | 20955 | 9.62 G |
| Ours | 2.74 G | 0.316 | 2685 | 18.1 G |
| Ours (with 3D reconstruction) | 4.66 G | 0.491 | 4512 | 23.1 G |

## K.2 VIRTUAL TRY-ON

We provide comparison results on the VITON-HD testing set in Figure 22. Zoom-in insets of the waistline highlight that our method maintains boundary fidelity under occlusions and nonrigid deformations, whereas competing methods exhibit seam breakage, color bleeding and ghosting artifacts.

## K.3 IMAGE RECOMPOSITING

We provide additional comparison results on the LVIS validation set in Figure 23.

## L LLM USAGE

We used ChatGPT 5 solely as a general-purpose writing assistant for minor phrasing as well as grammar and spelling corrections. The LLM did not contribute to research ideation, dataset design, model architecture, experiments, analyses, or conclusions, and it was not used to generate code,

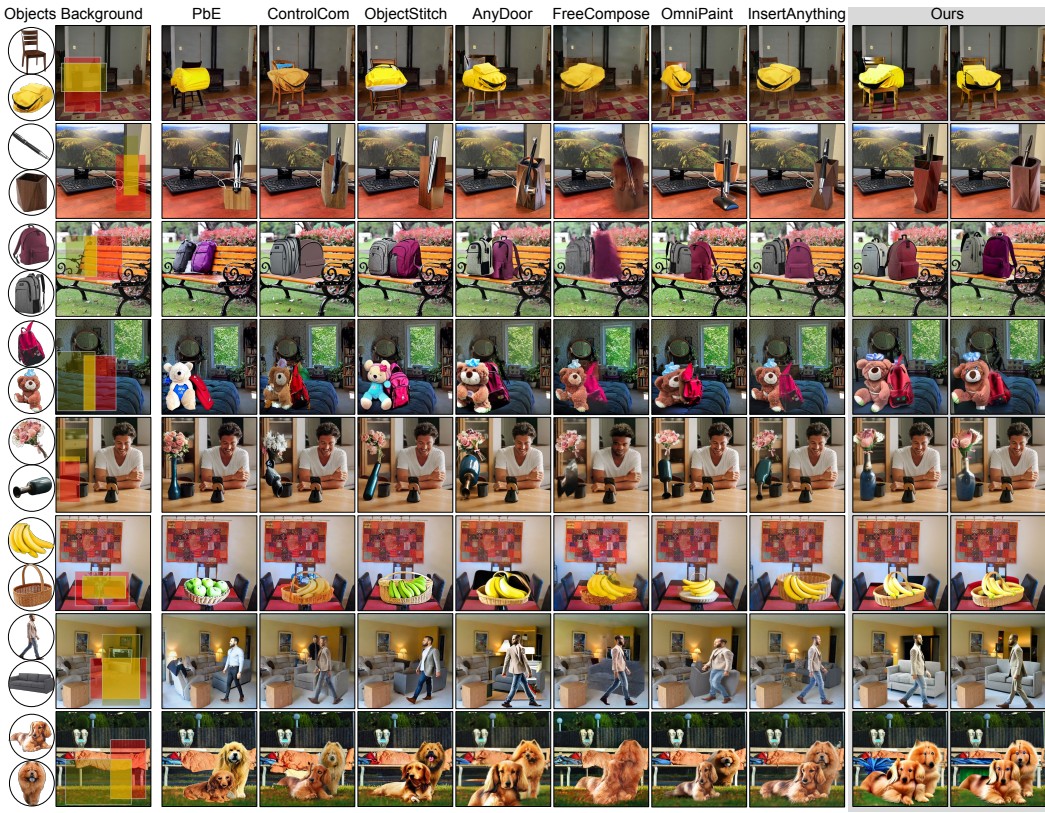

Figure 21: Qualitative comparisons for in-the-wild compositing, highlighting spatial relations: support, containment, occlusion, and deformation, corresponding to the teaser figure.

Table 7: Typical artifacts observed in the pairwise compositing results of Figure 21.

| Sample | Paint-by-Example | ControlCom | ObjectStitch | AnyDoor |
|---|---|---|---|---|
| 1 | Seat surface missing | Seat distorted, fused with backpack | No physical placement supported | Seat surface disappeared |
| 2 | Holder shortened, pen style changed | Pen attached on side of holder | Pen attached on side of holder | Pen attached on side of holder |
| 3 | Artifacts between backpacks | Back one occludes front one | Right backpack shows left backpack's color | Back one occludes front one |
| 4 | Toy style changed, backpack deformed | Contact region distorted | Toy style changed, contact region unrealistic | Toy not physically touching blanket |
| 5 | Vase shape altered | Vase not placed upright | Vase not placed upright | Vase and flowers both tilted |
| 6 | Banana turned into apple, no basket shadow | Banana fused with basket front | Banana placement caused basket gap | Banana partly fused with basket |
| 7 | Features of both human and sofa not preserved | Strange artifacts in occluded sofa region | Human leg splits sofa | Both human and sofa shapes unrealistic |
| 8 | Distorted leg | Fair | Distorted leg | Fair |

figures, or results. All technical content, equations, and claims were written and verified by the authors, who accept full responsibility for the paper. No confidential data were shared with the LLM, and any suggested text was reviewed and revised by the authors. The LLM is not an author.

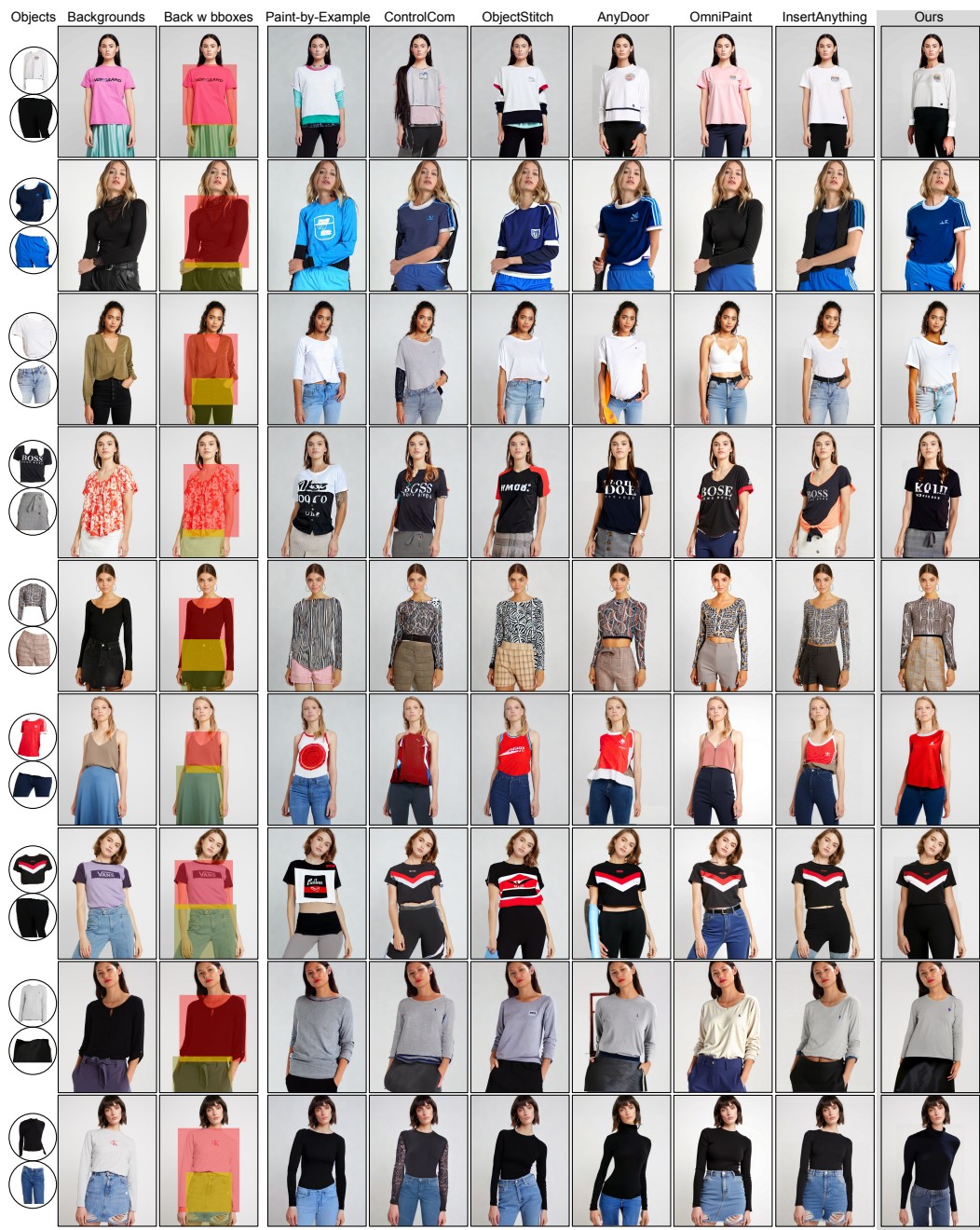

Figure 22: Qualitative comparison on the VITON-HD.

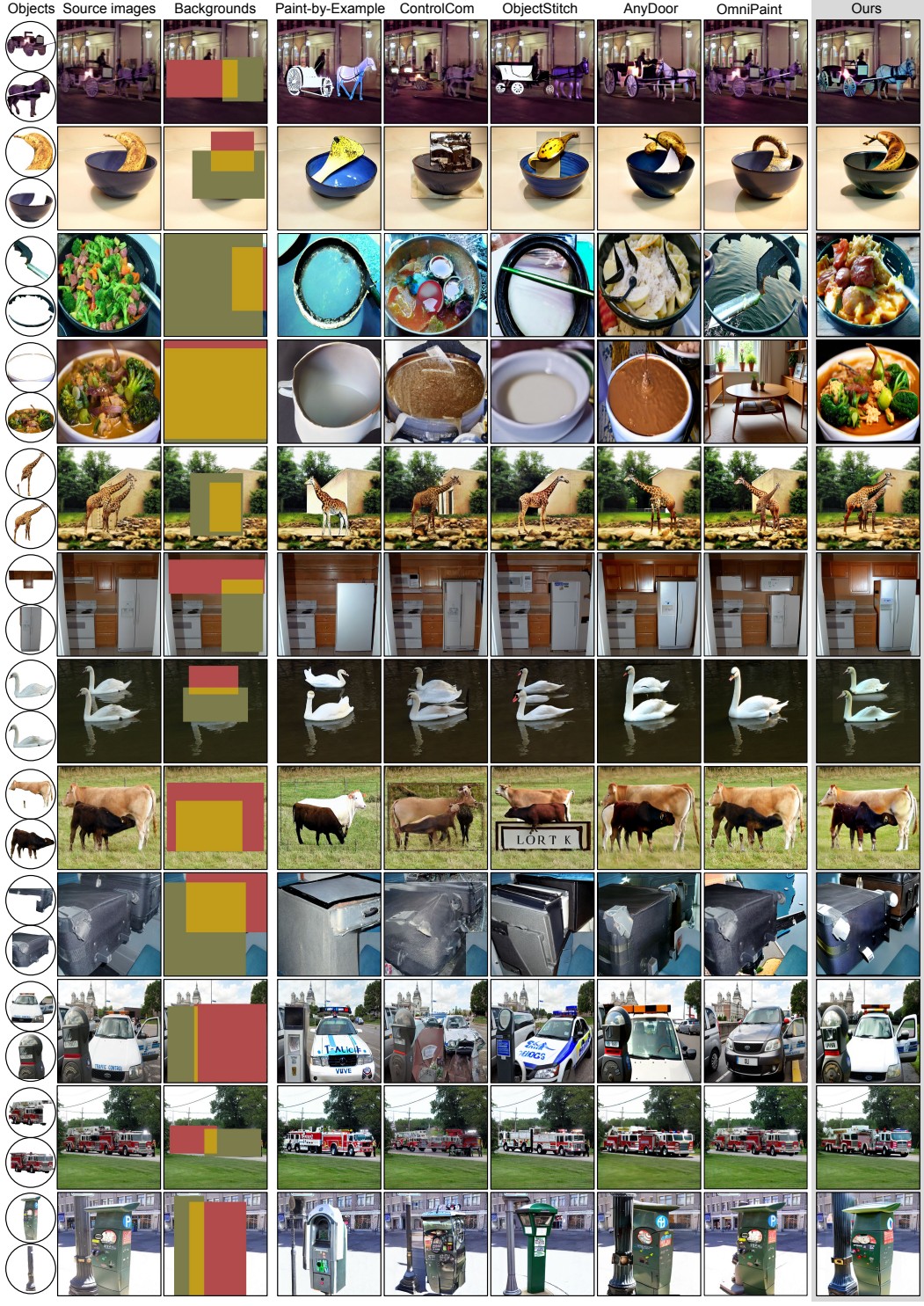

Figure 23: Additional qualitative comparison on the LVIS validation set.

