# OpenReview forum: "PICS: Pairwise Image Compositing with Spatial Interactions"
_ICLR.cc/2026/Conference — ICLR 2026 Poster_

### Official Review · Reviewer_z8Jf · 2025-10-28

**Soundness:** 3
**Presentation:** 3
**Contribution:** 3
**Rating:** 6
**Confidence:** 4

**Summary:**

The paper tackles the task of composing objects in parallel by explicitly modeling the compositional interactions among visible objects and background. The core is an Interaction Transformer that employs mask-guided Mixture-of-Experts to route background, exclusive, and overlap regions to dedicated experts. The proposed method outperforms baselines in inserting multiple objects.

**Strengths:**

1. I appreciate the task and the motivation, that sequentially insert objects into an image will overwrite previously generated content and inserting multiple objects is a meaningful task itself.
2. The visualized results seem good.
3. The paper is overall well-written.
4. I appreciate the interaction diffusion network as well as the interaction transformer block, that sounds reasonable.

**Weaknesses:**

1. It seems that most cases shown in the paper are inserting two objects. I wonder if there's any direct extension to multiple objects since previous methods can perform it sequentially. As the authors also claim in the limitation "restricted to two objects".
2. It seems that during training, the model takes in modal part (visible part) of an object? But I am assuming that during in-the-wild inference, the ideal input would be unoccluded instances like those shown in Fig. 5. Could the authors attach more examples as well as explaining how the proposed method can generalize well to this kind of setting?

**Questions:**

See above

---

> ### Author Response · Authors · 2025-11-26
>
> Thank you for your review. In addition to the general response, we address your specific comments below:
>
> **Visualization of unoccluded inputs:** In addition to the examples shown in Figures 5, 7, 8, 9, and 20, we further provide more qualitative results for unoccluded inputs in **Figure 21 (Appendix H)**. These results show that our method **generalizes well** to this setting: even when both input objects are fully visible, the model is still capable of producing **correct pairwise spatial relations while maintaining object identity** in the final composites.
>
> This generalization capability stems directly from our **overlap-expert design**. As illustrated in Figure 3(c), the overlap block is trained to synthesize the overlap region by **jointly leveraging the two partial object features**. Consequently, at inference time, when the inputs are intact rather than occluded, the same mechanism still applies: the overlap expert extracts the essential object-level features and learns to generate the appropriate interaction patterns (e.g., occlusion, containment, support). Because **intact inputs provide richer and cleaner cues than occluded training views**, the model naturally maintains strong robustness and compositional fidelity, consistent with the visual evidence shown in the additional figures.

---

### Official Review · Reviewer_M9rh · 2025-11-01

**Soundness:** 3
**Presentation:** 4
**Contribution:** 2
**Rating:** 4
**Confidence:** 5

**Summary:**

PICS is a diffusion-based framework for composing two objects into a background while maintaining spatial and visual consistency. The key idea is to perform parallel compositing of two objects rather than sequential editing. This paper introduces an Interaction Transformer with mask-guided experts and adaptive α-blending to handle overlapping regions and preserve boundaries, along with geometry-aware augmentations for robustness to pose changes.

**Strengths:**

The paper identifies a real issue, multi-turn compositing instability, and addresses it with a structured, parallel formulation. The task definition is well-motivated.

The mask-guided MoE and α-blending are simple but intuitive mechanisms to ensure boundary consistency.

**Weaknesses:**

It seems that the method is explicitly limited to pairwise (two objects) compositing.  Could PICS perform well on three-object, four object compositing? Please give me several visual examples if available.

Baselines are outdated. To the best of my knowledge, some open-source object insertion models can perform much better than the baselines you have chosen, such FreeCompose [1], OmniPaint [2], and Insert Anything [3]. Could you provide a comparison of these models?

The proposed mask-guided MoE and multiple cross-attention blocks likely increase inference cost. Could you report detailed runtime and parameter counts to quantify the overhead?

Figure 8 briefly shows degradation of compositing, but the causes are not analyzed in detail.

There is no visualization or strong justification for why geometry-aware augmentation improves compositing quality.



[1] FreeCompose: Generic Zero-Shot Image Composition with Diffusion Prior (ECCV 2024)

[2] OmniPaint: Mastering Object-Oriented Editing via Disentangled Insertion-Removal Inpainting (ICCV 2025, release time: 2025.03)

[3] Insert Anything: Image Insertion via In-Context Editing in DiT (arXiv 2025, release time: 2025.04)

**Questions:**

Refer to Weaknesses

---

> ### Author Response · Authors · 2025-11-26
>
> Thank you for your review. In addition to the general response, we address your specific comments below:
>
> **Comparison with FreeCompose, OmniPaint, and InsertAnything:** Thank you for pointing us to these additional baselines. We have included extensive visual comparisons in **Figures 2, 4, 21, 22, and 23**. From these results, we observe that existing methods often face challenges in maintaining **spatial consistency** and preserving fine-grained interaction details at object contact regions. For example, in Figure 2, none of the comparison models successfully capture the _bread-in-basket_ relation, suggesting that they have difficulty reasoning about fine-grained spatial compositing. In contrast, our method produces more coherent composites with improved spatial alignment and object-object interactions.
> We also provide quantitative comparisons in **Tables 1, 2, 3, and 6**. In the object recompositing setting, our model achieves the best performance across most metrics. In the object compositing setting, our method achieves superior FID and CLIP-score. Regarding DreamSim metric, AnyDoor and OmniPaint obtain higher scores: AnyDoor incorporates high-frequency object features as additional guidance, which enhances structure preservation at the expense of spatial harmony with the background; OmniPaint, in contrast, is built on a flow-matching FLUX backbone, whose stronger generative prior improves low-level fidelity but does not specifically target compositional consistency.
>
> **Computational overhead:** We have provided the computation cost in Table 6. Since our model **shares the same backbone with AnyDoor**, we can easily report and compare the overhead for the mask-guided MoE and multiple cross-attention blocks. Compared to AnyDoor (2.45 G parameters), our model (with 2.74 G parameters) has **slightly increased by 0.29 G parameters**. The inference time of our method is **0.316 min per sample**, slightly smaller than AnyDoor with 0.342 min per sample.
>
> **Visualization of 3D augmentation:** As suggested, we additionally show several visual comparisons of our model with and without 3D augmentation, as demonstrated in Appendix I and visualized in Figure 20. Incorporating 3D augmentation enables the model to synthesize a **broader range of viewpoint variations** for the composed objects. For instance, in the first row, the inserted donut exhibits clear geometric changes; in the second row, the flower is rendered from a novel orientation relative to its input view; and in the last row, the doll also appears under a noticeably different facing direction. These examples illustrate that coupling our model with 3D augmentation substantially improves viewpoint diversity, leading to richer and more flexible compositional generations.

---

### Official Review · Reviewer_r6im · 2025-11-01

**Soundness:** 3
**Presentation:** 3
**Contribution:** 3
**Rating:** 4
**Confidence:** 4

**Summary:**

This paper proposes PICS, a diffusion-based method for pairwise image compositing that shifts from sequential to parallel composition of two objects into a background, explicitly modeling spatial interactions like support, containment, occlusion, and deformation.
The main technical contribution includes an Interaction Transformer with mask-guided Mixture-of-Experts for region-specific processing (background/exclusive/overlap) and an adaptive alpha-blending mechanism for coherent overlaps, with geometry-aware augmentations for pose variations. The model is trained on ~1M samples from datasets including LVIS, Objects365, and VITON-HD. The experimental validation conducted by the authors claims the gains in compositing tasks on numerical scores, visuals, and user study feedbacks.

**Strengths:**

This paper offers a novel parallel compositing paradigm and MoE-based transformer that targets pairwise relations, addressing instability in multi-turn diffusion edits. The paper is well-written and easy-to-read. The experimental design is reasonable.

**Weaknesses:**

1. Despite superior metrics, visual fidelity remains unsatisfactory in some samples, with identity preservation issues such as altered stitch on the bag (2nd row, Figure 4), distorted bottle label (4th row, Figure 4), and changed shoe patterns (last row, Figure 5), suggesting limitations in retaining fine details during interaction modeling.

2. The restriction to exactly two objects may limit the method's broader applicability. Extensions to more than 2 objects are not explored in the paper, leaving scalability of the proposed method unclear.

3. While failure cases are shown, their analysis is brief, attributing problems to the shape encoder without detailed remedies or further ablations. Also, is this one of the possible reasons that lead to the poor visual fidelity mention in the weakness 1?

**Questions:**

1. Given observed fidelity losses in textures and labels (e.g., Figure 4 rows 2/4, Figure 5 last row), how will tuning the CFG scale or refining the shape encoder improve identity preservation without sacrificing harmony? Is there any other solution to improve visual fidelity?

2. How does the proposed Interaction Transformer extend to more than two objects? For instance, would the overlap expert generalize to multi-way intersections, or require modifications such as hierarchical routing?

3. In line 283, the authors mentioned they use the off-the-shelf single-view reconstruction model to render K auxiliary views as augmentation. Which model is used here? How does it impact inference time or robustness to extreme viewpoints?

I would be happy to raise my score if the author rebuttal addresses my concerns.

---

> ### Author Response · Authors · 2025-11-26
>
> Thank you for your review. In addition to the general response, we address your specific comments below:
>
> **Identity-preserving issue:**
> Thank you for the detailed observation. We agree that some fine-grained textures (e.g., stitches or small printed patterns) may change in a few cases. However, we would like to clarify that this issue is **not specific to our interaction modeling**, but a well-known limitation of current diffusion-based compositional pipelines.
> In our framework, object identity is primarily determined by the object encoder (DINOv2) and the scene decoder, rather than by the interaction module. The fine-detail changes pointed out by the reviewer occur **within the object region itself**, rather than at the interaction boundaries, which indicates that they originate from the encoder/decoder representation rather than the proposed interaction mechanism.
> This behavior is consistent with prior image compositing methods. For example, in some cases, even AnyDoor (which explicitly claims stronger identity preservation) still struggles to maintain high-frequency details across complex scenes.
> We appreciate the reviewer highlighting this direction, and improving fine-grained identity preservation is indeed an important next step that we plan to explore in future work.
>
>
> **Balance between identity preserving and harmony:** In Appendix D and Figure 15, we have already provided a comparison of composite images generated using different CFG scales. A higher CFG scale generally strengthens identity preservation, as the model is more strongly guided toward the object-specific features. However, this comes at the cost of reduced visual harmony and increased risk of saturation artifacts; lower CFG has the opposite effect. Thus, identity fidelity and compositional harmony form an intrinsic trade-off.
> In the future, several technical directions could further improve visual fidelity without destabilizing this balance. _(a) Stronger object encoders._ Enhancing the encoder’s ability to retain high-frequency details. For example, employing more expressive backbones such as DINOv3 may mitigate feature loss at the identity level. _(b) More capable generative backbones._ Using more powerful generative backbones such as flow matching based FLUX may improve the overall image generation quality. _(c) Improved guidance mechanisms._ Recent work [1] introduces alternative classifier-free guidance formulations that explicitly suppress oversaturation and may enable stronger fidelity without sacrificing harmony.
>
> **Discussion of the single-view reconstruction model:** We employ Zero123++ [2] (L900-L901) as our 3D reconstruction module. Table 6 reports the computational overhead introduced by this component. Using the reconstruction model increases the per-sample inference time from **0.316 min** to **0.491 min**. Given the improved viewpoint consistency it provides, this overhead is modest and remains within an acceptable range.
> We additionally show several visual comparisons of our model with and without 3D augmentation, as demonstrated in Appendix I and visualized in Figure 20. Incorporating 3D augmentation enables the model to synthesize a broader range of viewpoint variations for the composed objects. For instance, in the first row, the inserted donut exhibits clear geometric changes; in the second row, the flower is rendered from a novel orientation relative to its input view; and in the last row, the doll also appears under a noticeably different facing direction. These examples illustrate that coupling our model with 3D augmentation substantially improves viewpoint diversity, leading to richer and more flexible compositional generations.
>
> Regarding the question on extreme viewpoints, the performance of our method (Ours-with-3D-reconstruction) is **inherently coupled** with the quality of the underlying single-view reconstruction model. When the reconstruction model reliably recovers geometry under extreme viewpoints, our method naturally inherits this robustness. In practice, we observe that Zero123++ only on very rare occasions fails on challenging cases, producing coarse geometries whose shapes may **collapse toward cuboid-like forms**. Nevertheless, these reconstructions typically preserve the high-level texture and appearance of the object, which is the most critical cue for our compositing module. As a result, even when geometry degrades under extreme viewpoints, the preserved texture is sufficient for our method to maintain stable compositing behavior, and the impact on overall performance remains limited.
>
> [1] Eliminating Oversaturation and Artifacts of High Guidance Scales in Diffusion Models, ICLR’25
>
> [2] Zero123++: A single image to consistent multi-view diffusion base model. arXiv preprint arXiv:2310.15110, 2023

---

### Official Review · Reviewer_4pr4 · 2025-11-01

**Soundness:** 3
**Presentation:** 4
**Contribution:** 3
**Rating:** 6
**Confidence:** 3

**Summary:**

The authors developed a parallel image compositing framework, PICS, to resolve spatial inconsistencies in pairwise edits by explicitly modeling object interactions using a mask-guided Mixture-of-Experts. The paper is well-organized.

**Strengths:**

1. The paper is well-written, and the proposed methodology is sound.
2. The approach is well-supported by sufficient experimental results, including both quantitative and qualitative comparisons.
3. The core idea of the Mask-guided Mixture-of-Experts (MoE) is interesting.

**Weaknesses:**

1. Dependency on Mask Quality: The method's performance relies on precise input masks. The paper lacks a sensitivity analysis on mask quality (e.g., comparing results from high-quality vs. coarse masks or different segmentation sources), making it difficult to assess its robustness to imperfect inputs.
2. Background-based Gating: The gating mechanism for occlusion (Eqs. 6-9)  is primarily based on background-object similarity. This may lead to incorrect judgments when objects share similar textures or colors with the background. Incorporating direct object-object interactions might be necessary for more reliable occlusion reasoning.
3. Limited Physical Consistency: The method occasionally degrades the original object's geometry or texture , as seen in the failure cases (Figure 8).
4. High Computational Cost: Compared to existing methods, the model has a significant parameter count (2.74G – 4.66G), which could hinder practical adoption.
5. Scalability Limitation: The approach is restricted to pairwise compositing, making it less flexible for multi-object scenes.

**Questions:**

See previous.

---

> ### Author Response · Authors · 2025-11-26
>
> Thank you for your review. In addition to the general response, we address your specific comments below:
>
> **Effects on mask quality:**
> Thank you for pointing this out. For the mask of in-the-wild test samples, we use SAM to obtain the masks of objects. To assess the robustness of our method,  to segmentation masks of varying quality, we have added an additional experiment with three masks obtained by SAM, imperfect manually segmented masks and bounding box masks, as exemplified in Appendix F and Figure 17. Specifically, we compare pairwise compositing results using coarse masks including bounding boxes and manually annotated masks by (https://pixlab.io/annotate) against results obtained with high-quality SAM masks. Our findings show a clear trend: better segmentation masks lead to better compositing. In contrast, coarse or inaccurate masks tend to introduce undesired background cues from the object image, ultimately degrading the compositing quality.
>
> **Clarification about the gating query:**
> The reviewer’s concern appears to assume that the background query $q_g$​ encodes background **texture**, leading to appearance-based matching between background and objects. However, in our design, $q_g$ is **not** a texture descriptor. Because it is derived from the background feature $\mathbf{z}^{l-1}$, which is a deep semantic representation, the query encodes **learned spatial relationships**, including depth ordering, occlusion patterns, and object-background contact cues, not raw colors or textures. Thus, the gated scores in Equation (8) do **not** measure appearance similarity; instead, they reflect **how well each object’s aggregated representation explains the learned occlusion context at that location**.
>
> **Implicit object-object interaction:**
> Although the query originates from the background feature, the gating is **not** background-only. The softmax in Equation (8) jointly normalizes the two scores $(s _a, s _b)$, making the mixing weight $\alpha$ depend on the **relative alignment** between the two objects. Thus, each object directly influences the other’s contribution in Equation (9), forming an **implicit but effective object-object interaction**. This pairwise competition enables robust occlusion reasoning even when object and background textures are similar. We have added a brief explanation of these design choices in the main paper (Section 3.2, right after Equation (10)) to make this clearer to readers.
>
> **Physical consistency in failure cases:**
> We appreciate the reviewer’s observation. We would like to clarify that such cases do not originate from our compositing mechanism per se, but are a well-known issue shared by almost all image-based compositing methods. These failures are mainly due to **dataset bias and long-tail visual patterns**, rather than a fundamental limitation of our architecture. In particular, when the training data rarely contains certain background objects as entities that should be preserved, the model has limited supervision to distinguish them from editable background. Consequently, when a new object is composited nearby or partially overlaps these regions, the model may sometimes treat parts of such structures as background and slightly distort them, instead of perfectly preserving their geometry or texture. This observation is valid and points to an important research direction: developing image compositing models that can robustly preserve background object structure and ensure physically plausible interactions in scenes with complex background geometry remains an open challenge and is a valuable avenue for future work.
>
> **Computational cost:**
> Thank you for raising this point. As suggested by R-M9rh, we have added the computation costs of another 3 recent models in Table 6. We would like to clarify that the computational cost of our model is in a reasonable range compared with recent compositional or insertion-based diffusion models.
> Our parameter scale (2.7-4.6B) is substantially smaller than several models we added per reviewer M9rh’s request. For instance, OmniPaint-Flux contains **12.1B** parameters which is over **3x** larger, yet it runs comfortably on a single NVIDIA A6000 GPU. Our models also run inference on the same hardware without any memory constraints.
> The parameter range is also consistent with modern diffusion backbones (e.g., Stable Diffusion, FLUX), which commonly fall between 1B-8B (2GB-16GB). Thus, our model size is well within practical scales used in current generative systems.
> In summary, while our model is larger than lightweight baselines, it remains computationally feasible, smaller than several existing models. In future work, we plan to explore lightweight variants (e.g., parameter sharing across experts) to further improve efficiency without compromising quality.

---

### Author Response · Authors · 2025-11-26
**Summary of Changes**

We thank the reviewers for all of their feedback and comments on our paper. We have updated the paper with the following:

**Main changes:**

1. Experiment comparing FreeCompose (ECCV’24), OmniPaint (ICCV’25), InsertAnything (arXiv’25) in Figures 2,4,20,21,22, Tables 1,2,3,6. Thanks to **R-M9rh** for the suggestion.

&emsp;&emsp;a. Table 1: Pairwise object recompositing on LVIS dataset
| Method | mPSNR $\uparrow$ | mSSIM $\uparrow$ |mLPIPS $\downarrow$ | PSNR $\uparrow$ |FID $\downarrow$ |  LPIPS $\downarrow$ | CLIP-score $\uparrow$ | DINOV2-score $\uparrow$ | DreamSim $\downarrow$ |
|---|---|---|---|---|---|---|---|---|---|
| PbE (CVPR'23) | $10.24$ |  $0.4241$   | $0.4535$ | $15.29$ | $34.93$ | $0.4138$ | $81.42$  |  $0.4320$  | $0.4896$ |
|ControlCom (arXiv'23) | $11.82$ |  $0.3185$   |  $\underline{0.3986}$ | $\underline{17.61}$ | $26.93$ | $0.3375$ |  $\mathbf{85.39}$ |  $0.5264$  |   $0.3248$|
|ObjectStitch (CVPR'23) | $10.84$ |  $0.3471$   |  $0.4203$ | $16.55$ | $29.68$ | $0.3572$ | $85.01$ | $0.5574$   |   $0.3458$ |
|AnyDoor (CVPR'24) | $11.62$ |  $\underline{0.5283}$   |  $0.4185$ | $17.12$ | $27.17$ | $\underline{0.3302}$ | $84.99$ | $\mathbf{0.6089}$ |   $\underline{0.2820}$ |
|OmniPaint (ICCV'25) | $\underline{12.20}$ |  $0.3096$   |  $0.4618$ | $16.09$ | $\underline{26.25}$ | $0.3542$ | $83.11$ | $0.5673$ |   $0.2774$ |
|$\textbf{PICS (ours)}$ | $\mathbf{13.88}$ |  $\mathbf{0.5823}$   | $\mathbf{0.3221}$ | $\mathbf{18.27}$ | $\mathbf{24.99}$ | $\mathbf{0.2530}$ | $\underline{85.25}$ | $\underline{0.5713}$ |  $\mathbf{0.2659}$ |

&emsp;&emsp;b. Table 2: Pairwise object compositing on DreamBooth testing set
| Method |FID $\downarrow$ |  CLIP-score $\uparrow$ | DINOV2-score $\uparrow$ | DreamSim $\downarrow$ |
|---|---|---|---|---|
| PbE (CVPR'23) | $262.4$ | $51.95$  |  $0.2383$  |  $0.4321$|
|ControlCom (arXiv'23) | $273.4$  | $\underline{52.38}$ | $0.2414$ |  $0.3194$ |
|ObjectStitch (CVPR'23) | $\underline{260.4}$ | $51.35$ | $0.3203$  | $0.3374$|
|AnyDoor (CVPR'24) | $274.1$ | $51.24$ |   $0.3401$ | $\underline{0.2733}$|
|FreeCompose (ECCV'24) | $299.6$ | $51.71$ | $0.2157$ | $0.3521$|
|OmniPaint (ICCV'25) | $260.4$ | $50.32$ |   $\textbf{0.3741}$ | $\textbf{0.2632}$|
|InsertAnything (arXiv'25) | $266.0$ | $50.54$ | $0.3612$ | $0.2934$|
|$\textbf{PICS (ours)}$ | $\textbf{255.5}$ | $\textbf{54.02}$ | $\underline{0.3631}$ | $0.3054$|

&emsp;&emsp;c. Table 3: User study
| Method | Quality $\uparrow$ |Fidelity $\uparrow$ | Consistency $\uparrow$ |
|---|---|---|---|
|PbE | $5.13$ | $2.53$  | $8.70$ |
|ControlCom | $12.2$ | $15.2$  | $13.0$ |
|ObjectStitch | $12.8$ | $7.59$  | $15.9$ |
|AnyDoor | $14.1$ | $18.4$ | $12.3$ |
|FreeCompose | $2.56$ | $1.27$ | $4.35$ |
|OmniPaint | $17.3$ | $\textbf{19.0}$ | $10.9$ |
|InsertAnything | $16.0$ | $18.4$ | $12.3$ |
|$\textbf{PICS (ours)}$ | $\textbf{17.7}$ | $17.7$ | $\textbf{22.5}$ |

&emsp;&emsp;d. Table 6: Inference-time computation cost
| Methods | Params | Time (min) | GFLOPs | GPU Memory |
|---|---|---|---|---|
|PbE | 1.31 G   |   0.722    | 1048 |   8.70 G   |
|ControlCom  | 1.37 G  |   0.684   | 1123 |  21.5 G |
|ObjectStitch   | 1.31 G  |   0.241   | 852 |   7.04 G  |
|AnyDoor | 2.45 G  |   0.342  | 2450 | 15.8 G   |
|FreeCompose |  1.07 G |  1.912 | 678  | 5.79 G|
|OmniPaint | 12.1 G  | 1.516 | 33035 | 23.1 G   |
|InsertAnything  | 0.52 G  |  1.379 | 20955 | 9.62 G |
|$\textbf{Ours}$ | 2.74 G  | 0.316  | 2685 |  18.1 G |
|$\textbf{Ours (with 3D reconstruction)}$   | 4.66 G  |   0.491  | 4512 |  23.1 G |

Qualitatively, our model produces spatially more consistent compositions than all competing methods. For the recompositing tasks, it outperforms other approaches on most quantitative metrics. For the compositing tasks, OmniPaint achieves slightly higher DINOv2-score and DreamSim, while our method attains superior FID and CLIP-score, reflecting better realism and semantic alignment. In the user study, our results are preferred in both overall visual quality and spatial consistency.

2. Experiments on 3-object and 4-object compositing (Section 4.4 and Appendix C). We thank **all reviewers** for this valuable suggestion.
3. An in-depth discussion of the failure cases supported by experiments (Appendix G). Thanks **R-r6im** and **R-M9rh**.
4. Experiments on the choice of object masks (Appendix F). Thanks **R-4pr4**.
5. More visualization of unoccluded inputs (Appendix H). Thanks **R-z8Jf**.
6. Additional comparative visualization with and without 3D augmentation (Appendix I). Thanks **R-M9rh**.

**Other changes:**
1. Added Clarification about the gating query and implicit object-object interaction pointed out by **R-4pr4** (L280-285).
2. Additional discussion of quantitative comparison (L428-431).

We hope the reviewers will consider our revised draft and update their scores accordingly. Please let us know if you still have concerns or additional questions.

---

### Author Response · Authors · 2025-11-26
**General Response**

We thank the reviewers for their thoughtful feedback. We appreciate that all the reviewers found the problem formulation and motivation interesting (4pr4, r6im, M9rh and z8Jf), especially with the use of Mixture-of-Experts (MoE) and that our proposed approach is sound (r6im, M9rh, z8Jf). The reviewers found our paper was well-written (4pr4, r6im, z8Jf) and that we had extensive experiments and results for our approach (4pr4).

We provide responses to shared comments below and further direct responses to each respective review:

**Multi-object scenarios [4pr4, r6im, M9rh, z8Jf]:**
Thank you for the valuable suggestions. Our model can be naturally extended to multi-object compositing. For each object code $c _p$, $p\in \\{1, …, M\\}$, we first compute $\tilde{\mathbf{c}} _{p}$ by cross-attending each object code with the background gating query, which captures its location-specific alignment with the background. We then compute a compatibility score $s_p$ for every object code $c_p$, reflecting its spatial relationship with the background scene. The binary mixing weight in Equation (8) is generalized by replacing it with

$\alpha _p=\frac{\exp(s _p/\tau)}{\sum _{j=1}^{M}\exp(s _j/\tau)}, \qquad p=1,\dots,M.$

This softmax normalization jointly reasons over all objects. Finally, instead of the two-object $\alpha$-blending in Equation (9), we adopt a multi-object weighted fusion:

$\mathbf{c} _{1:M}=\sum _{p=1} ^{M}\alpha _p\, \tilde{\mathbf{c}} _{p}.$

This formulation enables the model to learn interactions among an arbitrary number of objects in a unified manner. We have added a detailed discussion of the multi-object extension in Appendix C.

We additionally train dedicated models for 3-object and 4-object compositing using samples constructed entirely from the LVIS dataset, with representative examples shown in Figure 7 and Figure 8. In the 3-object case, the generated composites exhibit coherent occlusion ordering and contact relations, and object identities remain well preserved even in regions where multiple masks intersect, indicating that the interaction module can distribute appearance features without collapsing fine details. The 4-object case introduces more entangled spatial configurations, including multi-level occlusions. The model remains stable under these challenging settings: as illustrated in the bottom example of Figure 8, the backpack which is almost fully occluded, is correctly omitted in the final composite, showing that the model adheres to learned visibility constraints rather than hallucinating unseen content. We have incorporated this discussion in Section 4.4 and Appendix C.

**Analysis of failure cases [r6im, M9rh]:** We acknowledge that our earlier explanation was brief. A more accurate analysis is as follows: _(a) Unseen styles or rare object types._ Some failure cases involve textures or materials rarely seen during training, leading to weaker reconstruction quality.
_(b) Limitations of the encoder._ Inadequate encoding of high-frequency appearance (e.g. dense patterns) may propagate to later generative compositing modules.
Additionally, for (b), we provide a more complete analysis below. Current image-based object compositing methods rely on different choices of image encoders for encoding object features:
| Encoder |Methods |
|---|---|
|CLIP|ControlCom, ObjectStitch|
|DINOv2|AnyDoor, OmniPaint, PICS (ours)|
|SigLIP [1] |InsertAnything |

As shown in our comparisons in Figure 21 and Figure 22, **DINOv2 consistently preserves finer textures than CLIP**, especially for textures with small-scale patterns. This observation aligns with the findings reported in AnyDoor paper (page 7, Table 3), which also concludes that DINOv2 outperforms CLIP for appearance preservation.
Additionally, to further clarify the source of fidelity degradation, we conducted a new ablation by replacing the object encoder with CLIP while keeping all other components identical. Models are separately trained on the LVIS dataset only for fair comparison. As shown in Figure 18, the comparison reveals a clear degradation in appearance preservation when using CLIP: the composed objects exhibit diminished fine-grained details, such as the subtle skin-tone differences between the two elephants, the striped patterns on the T-shirt, and the features of the motorcyclist. These observations highlight the importance of using a stronger encoder for capturing high-frequency object textures that are crucial for identity-preserving compositing. This ablation study is added to Appendix G.

[1] Sigmoid Loss for Language Image Pre-Training, ICCV’23

---

### Author Response · Authors · 2025-12-03
**To AC and All Reviewers**

We thank AC and all reviewers for the time and effort devoted to evaluating our submission. Due to the unexpected interruption of the discussion period, we regret not having the opportunity to receive potential follow-up responces from the reviewers.

In our response and the revised manuscript, we provide focused analyses and targeted experiments directly aligned with the reviewers' specific suggestions. These additions aim to improve clarity and presentation of the method. We believe the clarifications offered adequately resolve the raised concerns and more clearly articulate the strengths and assumptions of our approach.

We appreciate the AC's careful consideration and hope that our thorough and precise responses demonstrate the robustness of our work.

---

### Meta-Review · Area_Chair_51KU · 2025-12-13

**Summary:**

Strengths pointed out by reviewers:
* The paper addresses an important issue of image editing, specifically compositing multiple objects into the same image with potential occlusions.
* The reviewers appreciate an interesting mask-guided MoE idea, where the inputs are routed to their respective experts.

Reviewer’s key concerns:
* Compositing to compose more than two objects  (reviewer r6im)
* Model details: reliance on segmentation masks., reliable separation of foreground and background for gating, need for 3D augmentations (reviewer 4pr4, M9rh)
* The reviewer questions the background gating that might fails in certain circumstances, like objects being similar to the background  (reviewer 4pr4)
* High compute and memory costs (reviewer 4pr4)
* Comparison to more recent baselines, like FreeCompute and OmniPaint (reviewer M9rh).

**Reviewer Concerns:**

Addressed:
* Compositing more than 2 objects (reviewer r6im). Authors added a more general formulation how to extend the approach to 3-4 objects and provided the experiments.
* Additional baselines (reviewer M9rh). Authors added extensive ablations against these models during the rebuttal, including a human study.
* Memory and compute: Authors argue that the cost is similar to other diffusion backbones and provided a comprehensive comparison of the runtime, memory and flops required for inference.
* Mask dependency: the authors provided the sensitivity analysis to the quality of the segmentation masks. They demonstrate that the approach is not very sensitive to masks and works also with bounding boxes.
* Clarifications about background gating and occluded objects (reviewers z8Jf, 4pr4). The authors clarified that they use depth estimates to do background gating and overlap experts that use occlusion-agnostic features.
* Justifying 3D augmentations (reviewer M9rh). Authors added relevant ablations.


Unaddressed:
* Issues with visual fidelity in details like bottle labels (reviewer r6im) and background distortions (reviewer 4pr4). The authors acknowledged the limitation and suggested that stronger encoders might solve the issue in the future.

**Reviewer Scores:**

Assumed scores:
* Reviewer 4pr4 -- initial rating: 6 -- unchanged
* Reviewer r6im -- initial rating: 4 -- increase to 6
* Reviewer M9rh -- initial rating: 4 -- increase to 7
* Reviewer z8Jf -- initial rating: 6  -- unchanged

---

### Decision · Program_Chairs · 2026-01-26

Accept (Poster)